



# Carbon sequestration in different urban vegetation types in Southern Finland

Laura Thölix[1], Leif Backman[1], Minttu Havu[2,3], Esko Karvinen[1], Jesse Soininen[2], Justine Trémeau[1], Olli Nevalainen[1], Joyson Ahongshangbam[2], Leena Järvi[2,4], and Liisa Kulmala[1]

[1]Climate System Research, Finnish Meteorological Institute, Helsinki, Finland
[2]Institute for Atmospheric and Earth System Research (INAR), University of Helsinki, Finland
[3]Centre national de recherches météorologiques (CNRM), Université de Toulouse, Météo-France, CNRS, Toulouse, France
[4]Helsinki Institute of Sustainability Science (HELSUS), University of Helsinki, Finland

**Correspondence:** Laura Thölix (laura.tholix@fmi.fi)

**Abstract.** Many cities seek carbon neutrality and are therefore interested in the sinks of urban vegetation. However, the heterogeneous nature of urban vegetation and environmental conditions limit comprehensive measurement efforts setting expectations for carbon cycle modelling. In this study, we examined the performance of three models – JSBACH, LPJ-GUESS, and SUEWS – in estimating carbon sequestration rates in both irrigated and non-irrigated lawns, park trees (*Tilia cordata*), and urban forests
(*Betula pendula*) in Helsinki, Finland. The test data included observations of various environmental parameters and component fluxes such as soil moisture and temperature, sap flow, leaf area index, momentary photosynthesis, soil respiration, and net ecosystem exchange. Our analysis revealed that these models effectively simulated seasonal and annual variations, as well as the impacts of weather events on carbon fluxes and related factors. However, validating absolute flux levels proved challenging due to observational constraints, particularly concerning mature trees and that in urban areas net ecosystem exchange
measurements include some anthropogenic emissions. Irrigation emerged as a key factor often improving carbon sequestration while tree-covered areas demonstrated greater carbon sequestration rates compared with lawns on an annual scale. Notably, all models demonstrated similar mean net ecosystem exchange across a studied urban vegetation area on an annual scale over the study period. However, compared to JSBACH, LPJ-GUESS exhibited higher carbon sequestration rates in tree-covered areas but lower rates in grassland types. All models indicated notable year-to-year differences in annual sequestration rates, but since
the same factors, such as temperature and soil moisture, affect processes both assimilating and releasing carbon, connecting the years of high or low carbon sequestration to key meteorological means failed. Overall, this research emphasizes the importance of integrating diverse vegetation types and impacts of irrigation into urban carbon modelling efforts to inform sustainable urban planning and climate change mitigation strategies.

## 1 Introduction

The majority of combustion of fossil fuels occurs within urban environments, contributing to an estimated 40–50 % of global anthropogenic greenhouse gas emissions (Marcotullio et al., 2013). Cities worldwide are actively engaged in climate change mitigation efforts, formulating strategies to decrease their anthropogenic emissions (Rosenzweig et al., 2010; Reckien et al.,



2014; Mitchell et al., 2022) and striving for carbon neutrality. The pursuit of carbon neutrality has highlighted the role of urban
vegetation, perceived as a cost-effective method with minimal management requirements, for offsetting emissions. Moreover,
green spaces in urban areas offer diverse ecosystem services, such as mitigating excess water flows (Berland et al., 2017) and
enhancing human health and well-being (Wolf et al., 2020). The presence of green areas contributes to improved air quality
(Beckett et al., 2000; Tallis et al., 2011), alleviation of the urban heat island effect (Rahman et al., 2020), and provision of
recreational spaces.

As a result, there is an interest and a need to understand and quantify carbon storage and fluxes in urban green areas and
the impact of management practices on carbon. This information can further be used to guide urban planning efforts aimed at
mitigating climate change by carbon-smart design. Urban carbon dynamics are influenced by a wide range of factors, including
land use and the selection of species, the intensity of use and management, and microclimatic conditions. The heterogeneity of
the urban environment creates variation in the conditions for carbon sequestration, most importantly in temperature, humidity,
$CO_2$ concentration, and radiation, as well as in fertility and soil conditions (Bezyk et al., 2018). It is well-known that these local
effects modify the urban carbon fluxes by altering, for example, the length of the growing season (Imhoff et al., 2004; Wohlfahrt
et al., 2019), tree growth (Dahlhausen et al., 2018), and soil conditions (Edmondson et al., 2016). In addition to creating local
heterogeneity, these factors also vary from year to year and the variation will be even more pronounced under climate change,
since extreme climatic events are more frequent and last for longer periods (Kim et al., 2018). Hence, warming could reduce
tree growth (Meineke et al., 2016) and, remarkably, increase soil respiration (Rustad et al., 2000), further impacting inter-annual
variation in carbon dynamics.

The heterogeneity of urban green spaces, coupled with the year-to-year variability in weather patterns, poses challenges for
empirically estimating carbon sequestration. Consequently, ecosystem models are needed to quantify both the present extent
and the future development of carbon sequestration. Based on extensive empirical data in native and managed ecosystems,
many comprehensive models on photosynthesis (Mäkelä et al., 2004; Hari et al., 2017), plant internal carbon balance (Schiestl-
Aalto et al., 2019), net ecosystem exchange of forests (Lindeskog et al., 2021; Bergkvist et al., 2023) and agricultural fields
(Ma et al., 2023; Lindeskog et al., 2013), and land surface models such as JSBACH (Reick et al., 2013) and LPJ-GUESS
(Smith et al., 2014; Lindeskog et al., 2021) have been used during recent years, but these models are not developed and tested
for urban environments. In urban green spaces, the response of vegetation to environmental factors can be altered as there are
limitations in soil water availability, different soil types, higher evaporative demand caused by increased temperatures, altered
biological competition and interaction, heavy management, and species atypical for local native or agricultural vegetation.
Therefore, applying those models in urban green spaces includes larger uncertainty.

In addition to modeling efforts aimed at natural and rural ecosystems, specialized tools for carbon sequestration have been
employed to simulate urban green areas. For example, the Vegetation Photosynthesis and Respiration Model (VPRM) (Ma-
hadevan et al., 2008) has been utilized in various cities (Hardiman et al., 2017; Wang et al., 2021; Wei et al., 2022), as well as
providing initial estimates for inversion models (Lian et al., 2023). However, the model relies on remote sensing data related to
vegetation and lacks internal phenological modules, which limits its suitability for simulating future scenarios. Another com-
monly used tool is the GIS-based i-Tree software (Nowak and Crane, 2000), which estimates carbon sequestration and storage



using biomass equations (McPherson et al., 2005, 2011; Soares et al., 2011; Russo et al., 2014). However, i-Tree is subject to certain constraints as it cannot adequately account for local conditions and climatic variations, provides limited descriptions of vegetation other than trees, and does not consider the impact of soil. The urban land surface model SUEWS (Surface Urban Energy and Water Balance Scheme, Järvi et al., 2011) has been used to estimate the $CO_2$ flux for example in Helsinki (Havu et al., 2024) and Beijing (Zheng et al., 2023). Model validation is primarily reliant on eddy covariance (EC) data, which typically integrates $CO_2$ fluxes across heterogeneous surfaces and various vegetation types. Consequently, this approach may not fully capture certain habitat-specific characteristics.

The aim of this study is to estimate the applicability of different ecosystem models with varying features and complexity to describe the carbon cycle in diverse urban vegetation. For this purpose, we analyzed the model performance on different carbon cycle components and their driving and limiting environmental factors utilizing observational data collected in various common urban vegetation types in Helsinki, Finland. We addressed three specific research questions: 1) do different vegetation types significantly vary in the carbon sequestration rates, 2) do we see a notable year-to-year variation in the annual carbon sequestration, and 3) how well the different models are able to depict variations in carbon sequestration?

## 2 Materials and Methods

### 2.1 Study area

The study took place in Kumpula, Helsinki, Southern Finland (60°12'11.3" N 24°57'40.4" E, Fig. 1fg). The area belongs to the humid continental climate class (Dfb) in Köppen and Geiger classification (Kottek et al., 2006), as the average temperature exceeds 10° C from June to September. Helsinki is located on the coast of the Gulf of Finland which cools the climate in spring and early summer, whereas in autumn and early winter the sea has a warming effect on the climate. In Helsinki, the mean annual temperature is 6.5° C and precipitation 653 mm (Jokinen et al., 2021).

The study was carried out within a vegetated sector situated southwest of the ICOS (Integrated Carbon Observation System) EC tower FI-Kmp, also known as SMEAR III (Vesala et al., 2008), (Fig. 1a). The study area is a 140° wide sector with a radius of 600 m from the tower. This sector encompasses both the Kumpula Botanic Garden and an allotment garden. It comprises various land cover types: 27 % paved surfaces, 8 % buildings, 27 % tree cover, and 38 % grass surfaces. In 2020–2021, an intensive measurement campaign took place over four sites located within the vegetated sector: a park area with trees and a partly irrigated open lawn, a non-irrigated open lawn and an urban forest (Fig. 1b–e). Details of the soil properties at the sites are given in Table S1.

The urban forest (60°12'07.7" N 24°57'33.0" E) dominated by silver birches (*Betula pendula*) is a rather small (25 m × 30 m) area located next to the parking area at the eastern side of the Kumpula Botanic Garden (Fig. 1b). Trees were planted around 1990. In 2021, the dominant birches were approx. 23.6 cm in diameter at breast height and 22 m in height. The vegetation was not actively managed and *Aegopodium podagraria* prevailed in the ground layer on the site. The soil was classified as sandy loam. The forest was surrounded by an open managed meadow on its northwestern side and by another forest on the east side with *Betula pubescens*, *Alnus glutinosa*, *Ulmus glabra*, *Acer platanoides*, and *Populus tremula*.



The park area (60°12'08.4" N 24°57'21.4" E), in the Kumpula Botanic Garden of the Finnish Museum of Natural History, included four linden (*Tilia cordata*) trees (Fig. 1c, hereafter called park trees) and irrigated lawn (Fig. 1d). The linden trees were originally planted in 1989 and transplanted in 1995 to their current locations close to a gravel footpath. In 2021, the average diameter at breast height was 26.3 cm and the height 12.5 m. The ground vegetation below the trees was mowed a few

times each year. The ground vegetation was rather diverse, including common grass species (*Poa pratensis*, *Lolium perenne*, *Festuca* sp., *Phleum pratense*, *Dactylis glomerata*, and *Alopecurus pratensis*) and also some more or less rare forbs, such as *Cerastium fontanum* and *Dianthus deltoides*.

In the open lawn area, next to the park trees, the vegetation was dominated by *Poa pratensis*, some other common lawn species including *Lolium perenne*, *Polygonum aviculare*, *Trifolium repens*, and *Achillea millefolium* were also present. The

lawn was fertilized once every few years; most recently in the spring of 2021. An irrigation system with sprinklers was manually regulated to work in conditions that supposedly featured low soil moisture. However, some small patches on the lawn area were also poorly irrigated. Mowing of the lawn was done by a robotic lawnmower, and the grass clippings were shredded and left on the lawn.

Last, the non-irrigated lawn (60°12'14.0" N 24°57'22.2" E) was surrounded by three footpaths, 150 m away from the

irrigated lawn (Fig. 1e). The eastern side of the lawn was bordered by *Acer platanoides*, *Populus tremula*, and *Lonicera* sp. shrubs. The ground vegetation at this site was dominated by a mixture of *Poa pratensis*, *Lolium perenne*, and *Festuca* sp., coupled with *Polygonum aviculare*, *Trifolium repens*, *Achillea millefolium*, *Plantago major*, and *Taraxacum* sp. The soil texture for the non-irrigated lawn was assumed to be the same as for the park site due to the close distance.

## 2.2   Models

### 2.2.1   JSBACH

JSBACH (Reick et al., 2013), the land component in Earth system models developed by the Max-Planck Institute for Meteorology (Giorgetta et al., 2013), simulates terrestrial energy, hydrology, and carbon fluxes through a suite of submodels. Vegetation is represented by plant functional types (PFT), with this study focusing on C3 grasses and broadleaved deciduous tree PFTs.

Photosynthesis of C3 plants in JSBACH follows the model by Farquhar et al. (1980). The photosynthesis is calculated once

to get the unstressed canopy conductance, which is then scaled based on the soil moisture in the root zone to get the canopy conductance and photosynthesis under water stress. The soil hydrology parameters are set according to soil texture following Hagemann and Stacke (2014), and the hydrology is simulated with a multilayer soil hydrology scheme.

Seasonal leaf area index (LAI) development is described by the Logistic Growth Phenology (LoGro-P) model (Böttcher et al., 2016). In the case of broadleaved deciduous trees this development depends on air temperature. The grass phenology

additionally incorporates soil net primary productivity as a determining factor. The maximum LAI was set based on Sentinel-2 data (see later). In addition, the phenology parameters were adjusted to improve the agreement with Sentinel-2 observations regarding the temporal development of LAI.



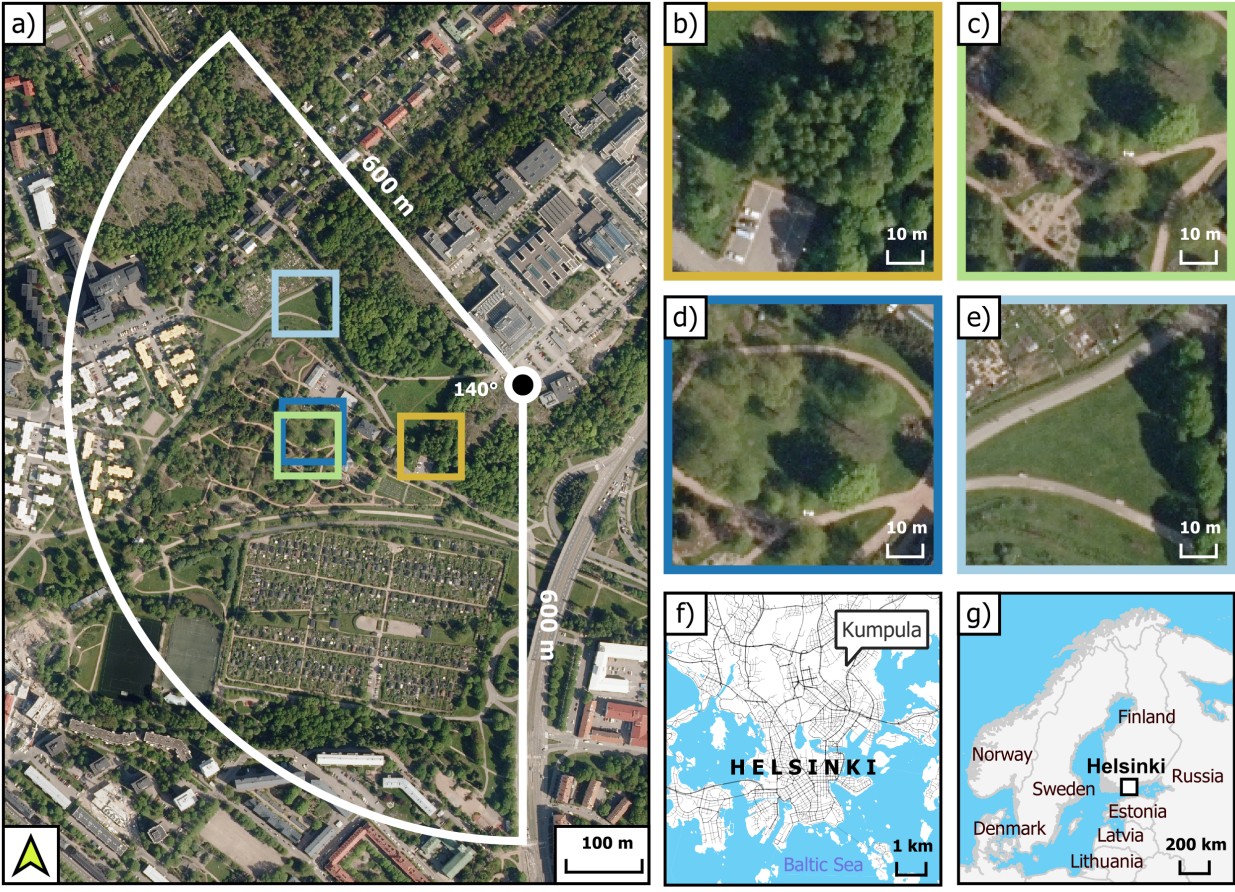

**Figure 1.** a) The studied vegetation types (coloured squares) and the modelling target area of this study (white-lined sector), that is, the vegetation sector described in Järvi et al. (2009) roughly indicates the footprint of the eddy covariance measurements (the tower is represented as a circle). The focused images show the different vegetation types studied: urban forest (b), the park with linden trees (c) and irrigated lawn with some non-irrigated patches (d), and non-irrigated lawn (e). The maps show the location of Kumpula in Helsinki (f) and the location of Helsinki and Finland (g). Orthophotos by National Land Survey of Finland (2023a), background maps built with topographic database by National Land Survey of Finland (2023b) and global administrative borders by GADM (2023).

The dynamics of litter and soil carbon in JSBACH are modeled based on the Yasso soil carbon model (Goll et al., 2015) that features five carbon pools reflecting the chemical quality of organic matter. The model distinguishes between woody and 125 non-woody organic material. The litter pools are further divided into above- and belowground pools, totaling 18 carbon pools. These pools accumulate carbon from litter flux and root exudates. Decomposition transfers carbon between pools and to the atmosphere.



The model was driven with hourly observation-based data of air temperature, precipitation, shortwave and longwave radiation, relative humidity, and wind speed. The simulations include an 8000-year spin-up period to establish the soil carbon pools.

The final simulations cover the period from 2005 to 2021. The forest (birch), park (linden), and lawn sites were simulated separately. The park and lawn sites were also simulated with irrigation (see Sect. 2.4).

### 2.2.2 LPJ-GUESS

Lund–Potsdam–Jena General Ecosystem Simulator (LPJ-GUESS, Smith et al., 2001, 2014; Lindeskog et al., 2021) is a process-based dynamic global vegetation terrestrial ecosystem model (DGVM) designed for regional and global studies. It

includes a detailed representation of forest ecosystem composition and stand dynamics. It simulates the dynamics and composition of vegetation in response to changes in climate, land-use change, atmospheric $CO_2$, and nitrogen. LPJ-GUESS simulates potential vegetation as a mixture of about 20 PFTs which compete with each other for light, space, and soil resources in each simulated grid cell. Each PFT is characterized by growth form, phenology, photosynthetic pathway (C3 or C4), bioclimatic limits for establishment and survival, and, for woody PFTs, allometry and life history strategy. All individuals of a given age

cohort are assumed identical (Knorr et al., 2016). Primary production and plant growth follow the approach of LPJ-DGVM (Sitch et al., 2003). The plant physiological processes of photosynthesis, respiration, stomatal regulation, and phenological development are simulated at a daily time step within the ecosystem. Soil hydrology and plant water uptake is modelled using a two-layer soil profile. LPJ-GUESS includes dynamic cycling of soil carbon based on the CENTURY model (Parton et al., 1993). Population dynamics (recruitment and mortality) are represented as stochastic processes, influenced by current resource

status, demography, and the life history characteristics of each PFT (Hickler et al., 2012).

All simulations in this study were initialized with a 500-year spin-up using atmospheric $CO_2$ concentration from 1901 and repeating detrended ERA5-land data from 1951 to 1980. The actual simulations covered the years 1901–2021. In this study, separate simulations for each vegetation type were run (forest, non-irrigated park, irrigated park, non-irrigated lawn, and irrigated lawn). The results of the whole vegetated sector were calculated from the separate simulations. In the urban forest

simulation, the birches were planted in 1990 and the tree density was set to 5000 trees $\text{ha}^{-1}$. In the park, the sparse linden trees were planted in 1988 and the density was set to 600 trees $\text{ha}^{-1}$. Only C3 grass was set to grow in the lawn simulations. The irrigation rates used in the simulations are introduced in Section 2.4.

The LAI calculations in autumn were changed from the standard LPJ-GUESS version. Originally LAI stayed at its maximum value until the growing season had lasted 210 days. We shortened the growing season length to 190 days to better fit the

circumstances in Helsinki and also decreased LAI by 1.5 % (trees) or 2 % (lawn) in a day when the daylight hours were less than 14 hours and soil temperature was less than 10° C. Maximum lawn LAI was limited to an average of 2.5 $\text{m}^2 \text{m}^{-2}$ and the previous year's maximum LAI for simulating the lawn mowing. Also, the LAI of the lawn during dry periods was changed from the LPJ-GUESS standard version to better correspond with the observations: In this study, LAI was decreased by 8 % after running out of water supply, and increased by 8 % after refilling.



### 2.2.3 SUEWS

The Surface Urban Energy and Water Balance Scheme (SUEWS, Järvi et al., 2011, 2019) is a neighbourhood scale urban land surface model to simulate the urban surface energy, water, and $CO_2$ balances. The urban surface is separated into seven connected surface types (buildings, paved surfaces, grass, evergreen trees and shrubs, deciduous trees and shrubs, bare soil, and water), with each surface, except water, having a single soil layer beneath. The urban water balance considers evapotranspiration simulated with the Penman-Monteith equation modified for urban environments (Grimmond and Oke, 1991), irrigation with either manual or automatic sprinklers, and runoff between the surfaces (Järvi et al., 2011). Canopy conductance is controlled by environmental factors, that is, LAI, solar radiation, air temperature, air humidity, and soil moisture, connecting transpiration to $CO_2$ uptake estimates, as in SUEWS the maximum potential photosynthesis depends on these environmental limiting factors (Järvi et al., 2019). Respiration is estimated with its exponential relation to air temperature. Both photosynthesis and respiration can be simulated separately for grass and trees; however, the model output is the mean value from the whole simulation domain. In SUEWS, the local 2 m air temperature is simulated by taking into account urban effects (Tang et al., 2021) and is used to estimate photosynthesis and respiration.

The model parameters are based on Järvi et al. (2019). The daily LAI is simulated by accounting for its dependence on growing degree days (GDD) and senescence degree days (SDD), both influenced by air temperature. Here, we optimized the rates of leaf-on and leaf-off with satellite-based LAI (see Sect. 2.5.1) to expedite the attainment of maximum LAI while ensuring a more gradual decline during leaf-off. For high latitudes, SUEWS incorporates day length and mean daily temperature as limiting factors for the initiation of senescence. The previously utilized value of 12 h (Järvi et al., 2014) leads to a delay in the onset of leaf-off. Therefore, we adjusted the threshold day length to 14 hours (Havu et al., 2024).

As SUEWS is an urban land surface model, the model runs were made only for the whole vegetated sector. The model run was made with hourly resolution for the whole period 2005–2021 with 2005 used as a spin-up year. Two different simulations were made to enable comparison between different observational sites: one with irrigation and one without irrigation.

### 2.3 Driver data

Two sets of hourly driver data were prepared: one for JSBACH and LPJ-GUESS mainly based on observations near ground level, and another for SUEWS mainly derived from tower data recorded at 32-meter height. Separate forcing data were used for SUEWS, given its characterization as an urban land surface model designed to simulate local climate conditions utilizing forcing data obtained in the inertial sublayer (Tang et al., 2021).

The first set of driver data included air temperature (2 m), precipitation (1.5 m), shortwave (2 m) and longwave (32 m) radiation, relative humidity (2 m), air pressure (1.5 m), and wind speed (10 m). The primary data source was observations from the Kumpula weather observation station (60°12'11.1"N 24°57'40.7"E), operated by the Finnish Meteorological Institute (FMI, 2022). Data gaps were filled using observations from the urban measurement station SMEAR III, located next to the weather station (Järvi et al., 2009; SMEARIII, 2022). The second set of driver data included air temperature, shortwave and longwave radiation, and wind speed from the SMEAR III measurement tower at 32 m height, alongside precipitation, relative



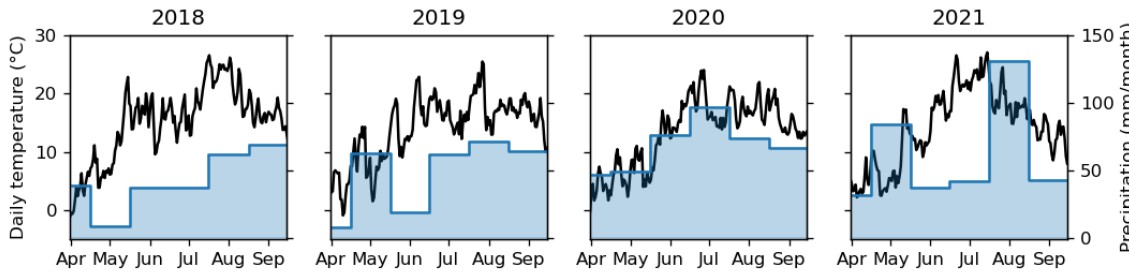

**Figure 2.** Observed daily temperature and monthly precipitation in Kumpula during 2018–2021.

humidity, and air pressure from the nearby rooftop at 26 m. Both observational datasets were gap filled by hourly ERA5-Land data (Muñoz Sabater et al., 2021; Muñoz Sabater, 2019). ERA5-Land data was used from 1951 to 2004. Data for model spin-up

in the years 1900 to 1950 was randomly generated based upon the years from 1951 to 1980.

The performance of the models was evaluated during two study periods. The primary and most intensive one covered the years 2018–2021, during which 2018 exhibited the highest temperatures and the least precipitation, while the subsequent years were more similar to each other (Fig. 2). According to the driver data, the growing degree days to base $5°$ C (GDD5) for the years 2018, 2019, 2020, and 2021 were $1964°$ C, $1679°$ C, $1720°$ C and $1777°$ C, respectively. The accumulated precipitation

during the irrigation period (May–August) was 144 mm, 215 mm, 296 mm and 293 mm for the corresponding years. The secondary study period was longer covering the years 2006–2021 (Figure S1). During the period, 2017 had the lowest and 2018 the highest growing degree days and maximum temperatures (Table S2). 2006 and 2011 also had GDD5 above $1800°$ C. The years 2006 and 2018 had the lowest amounts of summertime precipitation and the highest number of dry days (Table S2), whereas 2007 and 2009 had over 30 % more rainfall during summertime than average.

**2.4 Irrigation**

The irrigation scheme in SUEWS (Järvi et al., 2011) considers both automatic and manual irrigation on an hourly scale, as they tend to differ in their timing and response to weather conditions. The daily water use ($I_e$, $\mathrm{mm\,day^{-1}}$) is estimated from mean daily air temperature ($T_d$) and time since rain ($t_r$, day):

$$I_e = b_0 + b_1 T_d + b_2 t_r, \tag{1}$$

where $b_0, b_1$, and $b_2$ are site-specific coefficients. The daily precipitation is then divided to the the fraction of irrigated surfaces separately for each surface type specified in the model input files. The timing of the start and end of the irrigation season needs to be specified. For Helsinki, the irrigation season has been specified to start at the beginning of June and continue until the end of August. The site-specific coefficients $b_0, b_1$, and $b_2$ for Kumpula are $-5.8$ mm, $0.7$ $\mathrm{mm\,K^{-1}}$, and $0.2$ $\mathrm{mm\,day^{-1}}$, respectively (Järvi et al., 2017).





**Table 1.** Total water consumption between June and August as reported by the Kumpula Botanical Garden and corresponding estimate of daily average irrigation ($\mathrm{mm\,day^{-1}}$). The irrigation estimate based on an algorithm accounting for both temperature and precipitation, is given in the last column.

| Year | Total water consumption | Irrigation estimated from water consumption | Irrigation estimated by algorithm (used in JSBACH and LPJ-GUESS) |
|------|-------------------------|---------------------------------------------|-------------------------------------------------------------------|
|      | $\mathrm{m^3}$ | $\mathrm{mm\,day^{-1}}$ | $\mathrm{mm\,day^{-1}}$ |
| 2019 | 6471 | 1.21 | 1.32 |
| 2020 | 4726 | 0.74 | 0.71 |
| 2021 | 10997 | 2.44 | 2.34 |

There is no irrigation scheme in JSBACH and LPJ-GUESS, therefore the irrigation was included as extra precipitation during the summer months. The daily average irrigation for the irrigated areas in the studied vegetation sector was estimated from June to August water consumption data obtained from the Kumpula Botanic Garden. Water consumption data were available for 2019–2021 (Table 1). It was estimated that 2000 $\mathrm{m^3}$ was used annually for maintaining a pond in the garden, and this amount was removed from the total consumption. The area of the garden is 6 ha and we assumed that just 2/3 of this area was irrigated excluding, for example, maintenance and forested areas. Based on these assumptions we estimated the daily average irrigation from June to August to be 1.21, 0.74, and 2.44 $\mathrm{mm\,day^{-1}}$ in 2019, 2020, and 2021, respectively (Table 1). In order to extend the irrigation data to the whole simulated period 2005–2021, we used an algorithm that considered both temperature and precipitation. We used two-week running means; if the average temperature over two weeks was above 19° C (hot) or if the average precipitation was below 1.4 $\mathrm{mm\,day^{-1}}$ (dry), then the irrigation was 1.7 $\mathrm{mm\,day^{-1}}$. The extra precipitation was added to the hourly forcing data between 8 AM and 4 PM local standard time. If it was both dry and hot over two weeks then the irrigation was increased to 5.0 $\mathrm{mm\,day^{-1}}$. In this case the excess precipitation was added to all hours of the day. The algorithm produced year-to-year variation similar to that estimated from the garden water consumption data (Table 1). In the simulations with JSBACH and LPJ-GUESS, irrigation was added to the precipitation between June and August.

The irrigated area within the sector was estimated by assuming that 2/3 of both the Kumpula Botanic Garden and the allotment garden were irrigated, while areas outside of these gardens were not irrigated. As a result, we assumed that 28 % of the trees and 42 % of the grass areas (lawns) were irrigated in this sector.

## 2.5 Observations

### 2.5.1 Leaf area index

The four intensive sites were monitored using remote sensing imagery from the European Space Agency (ESA) Sentinel-2 satellites during 2018–2021. Atmospherically corrected Level-2A (L2A) Sentinel-2 multispectral data were retrieved using the GEE (https://earthengine.google.com/) cloud data platform. The scene classification band available in L2A products was used to filter away image acquisition dates during which the field is covered by snow, cloud, or cloud shadow. From the Sentinel-2



data, we calculated the leaf area index estimated using the ESA Sentinel Application Platform (SNAP) Biophysical Processor neural network algorithm (Weiss and Baret, 2016; Nevalainen et al., 2022). The uncertainty calculations are described in detail
by Nevalainen et al. (2022).

### 2.5.2 Photosynthesis

Light responses of net ecosystem exchange of $CO_2$ (NEE) of the partly irrigated lawn (Fig. 1d) were determined by manual chamber measurements conducted at four fixed plots ($60\,\mathrm{cm} \times 60\,\mathrm{cm}$) every second week from June to September in 2021 and 2022. Additional, more infrequent measurements were also conducted in May and from October to December. The setup con-
sisted of a transparent chamber attached to a $CO_2$ and $H_2O$ analyser (Li-840A, LI-COR, Inc., Nebraska, USA), air temperature and humidity sensor (BME280, Bosch Sensortec GmbH, Reutlingen, Germany), and photosynthetically active radiation (PAR) sensor (PQS1, Kipp & Zonen, Delft, Netherlands). One light response included five separate chamber closures in different light intensities: the first one with the transparent unshadowed chamber, then covering the chamber with 1–3 layers of mesh fabric, and eventually measuring with an opaque cover. Details of the setup, the protocol, flux calculations, and the light response
curve fitting are described in Trémeau et al. (2024). Momentary photosynthesis (GPP) was estimated for each 30-min utilising automatically recorded mean PAR at SMEAR III and the estimated light response curves assuming that 1) NEE at zero light represents total ecosystem respiration (TER), 2) TER was independent of the light intensity during the chamber measurements, and 3) GPP=TER-NEE. Daily GPP was obtained by summing up momentary GPPs. Even though the lawn area was mainly irrigated, it included also inadequately irrigated small patches which allowed us to divide the observations into adequately and
poorly irrigated ones.

In the park, $CO_2$ exchange of three separate linden shoots was measured with an automatic chamber system operating continuously at the site from June to September in 2020–2021. The chambers were mostly open enabling natural radiation and precipitation conditions. They closed automatically two times per hour for 2 minutes at a time resulting in 48 exchange rates per chamber in one day in naturally varying environmental conditions. The setup at each shoot consisted of a transparent chamber
of approximately 1 L, sample tubing, $CO_2$ analyser (GMP343, Vaisala Oyj, Vantaa, Finland), fan mixing the air, PAR (SQ-420X Smart Quantum Sensor, Apogee, Logan, USA) and RH/T (HygroClip 2 HC2A, Rotronic AG, Bassersdorf, Switzerland) sensors, and a pump that circulated sample air from the chamber to the analyzer and back in a closed loop. The $CO_2$ exchange was calculated from the rate of change in the $CO_2$ concentration during the closure. A simple light response curve was fitted into these exchange rates. Since the specific placement of the chambers may not have accurately reflected the typical light
conditions experienced by all leaves, we assessed instantaneous photosynthesis under unshaded conditions resembling those found in the uppermost canopy. This was achieved by utilizing the light response curve and photosynthetically active radiation (PAR) measurements obtained at the SMEAR III site. Daily photosynthesis was then calculated by summing the instantaneous rates, as described previously.

In addition to the automatic measurements conducted on the lindens, the light responses of $CO_2$ exchange of tree leaves were
manually measured using a portable gas exchange system (Walz GFS-3000, Heinz Walz, GmbH, Germany) with a standard measuring head (2 cm x 4 cm) on the trees in the park and urban forest during the summers of 2020 and 2021 at approximately





four-week intervals. During each measurement, $CO_2$ concentration was set to a fixed value of 415 ppm but the temperature was not set to any value and was following the ambient conditions. The measurement included different light intensities changing automatically from 1500 down to $0\,\mu\mathrm{mol\,m^{-2}\,s^{-1}}$ over 43 minutes. The measurements were performed on a healthy single leaf at two or three heights in three individual trees in the forest and linden sites on the southern or southwestern side of each tree. As a result, the linden site included altogether 9 light responses (three *Tilia cordata* trees, three heights) and the forest site 6 (three *Betula pendula* trees, two heights) during each repetition. The details of the measurement protocol and fitting of the light response curve are described in detail by Ahongshangbam et al. (2023). Again, daily photosynthesis during the measurement days was derived utilizing the light response curves and the automatic PAR measurements at SMEAR III.

### 2.5.3 Soil respiration

Soil respiration $CO_2$ flux was measured with manual chamber measurements both in the park (under the park trees) and in the urban forest in 2020–2021. The measurements were conducted weekly in May–Sep and fortnightly or monthly in Oct–Dec. The measurement setup consisted of a small opaque chamber (volume = $0.007434\,\mathrm{m^3}$) equipped with a $CO_2$ probe (GMP343, Vaisala Oyj), RH/T sensor (HMP75, Vaisala Oyj), and a small battery-powered fan to ensure air circulation within the chamber. Measurement data from the sensors was stored on site in a handheld reading console (MI70, Vaisala Oyj). Eight steel frames were inserted a couple of centimeters into the soil at both measurement sites, and the measurement chamber was placed on top of each of the frames for 4–5 minutes while measuring the $CO_2$ concentration, relative humidity, and air temperature change inside the chamber. The soil respiration $CO_2$ flux was calculated from the rate of change in the $CO_2$ concentration during the closure. The chamber system and the flux calculation are described in detail by Pumpanen et al. (2015) and Karvinen et al. (2024).

### 2.5.4 Soil moisture and temperature

Soil moisture was measured manually with a soil profile probe (PR2 (SDI-12), Delta-T Devices, Cambridge, UK) and a handheld reading console (HH2, Delta-T Devices) both in the park and in the urban forest. Six fibreglass access tubes (ATS1, Delta-T Devices) were installed in the soil in 2020 and measured weekly during the main growing season in 2020–2022. The profile probe measures soil moisture at 10, 20, 30, and 40 cm depths. Manual soil temperature measurements were conducted with a handheld console setup (Pt100 and HH376, Omega Engineering Inc., Connecticut, USA) together with all manual chamber measurements.

### 2.5.5 Sap flow

Sap flow measurements were conducted using the Thermal Dissipation Probe (TDP; Granier, 1985) consisting of two thermocouple needles (20–30 mm long) where one needle acts as a heating probe and the other as a reference probe. The sap flow density is calculated from the temperature difference between the heated probe and the reference probe. We measured three dominant trees at a height of 1.3 m at the park site and at 2.0 m at the forest site with a vertical distance of 10 cm between the



heated and the reference probe. The details of the setup, flow calculations and data are described in detail by Ahongshangbam et al. (2023).

### 2.5.6 NEE from eddy covariance

The net ecosystem exchange (NEE) was estimated from the eddy covariance measurements located in SMEAR III station (Vesala et al., 2008; Järvi et al., 2009). The source area of the 31-m measurements tower varies depending on the meteorological conditions, as the fetch can be larger during stable winter conditions, up to one kilometer. East to the tower lies one of the main roads leading to the city centre with high traffic amounts. Southwest to the tower (180–320°) lies the Kumpula Botanic Garden and allotment gardens (the vegetation sector), where the closest road is located 800 m from the mast.

The wind components were measured with an ultrasonic anemometer (USA-1, Metek GmbH, Germany) and the $CO_2$ mixing ratios with two infrared gas analysers (closed-path analyser LI-7000 or open-path analyser LI-7500; LI-COR, Lincoln, NE, USA), depending on the year as the open-path analyser was installed in 2005, and the closed-path in 2014. The 60-min flux values were calculated following Nordbo et al. (2012). The analysed period started in 2006 and ended in 2021. Measurements coming from the vegetation sector based on wind direction were used in order to limit emissions from the road located east of the tower. As flux values are 60-min averages from the 10 Hz data, some traffic-related emissions might still be included in the estimation, as well as emissions from human metabolism. The daily values of NEE were calculated from the vegetation sector, with complete data coverage for a full day available for only 15 % of the days. Consequently, when individual data gaps were shorter than four hours, they were filled using linear interpolation. This process resulted in an increase in data coverage to 25 % over the 16-year period.

## 3 Results

### 3.1 Soil conditions

JSBACH and LPJ-GUESS displayed a rather consistent pattern in simulating root zone temperature, following the observations closely in both the forest (not shown) and the park trees (Fig. S2). The Pearson's correlation coefficients of soil temperature ranged between 0.93 and 0.97 for these models (Table 2). SUEWS does not simulate soil temperature and consequently was excluded from the comparison.

All models exhibited similar predictions for soil moisture dynamics within the root zone in the urban forest and park sites for 2021 (Fig.3). However, it is noteworthy that the soil volume represented varies in the different models. For instance, JSBACH considered a root depth of 0.45 m for park trees (linden) and 0.65 m for forest trees (birch). LPJ-GUESS accounted for the soil moisture of the root zone, with 60 % of birch (forest) roots in the uppermost 0.5 m and 40 % in the next 1 m layer, and 80 % and 20 % for the park trees (linden), respectively. In contrast, SUEWS simulated soil moisture across the whole simulation domain from an average depth of 11–35 cm, depending on the surface type, resulting in no differentiation between vegetation



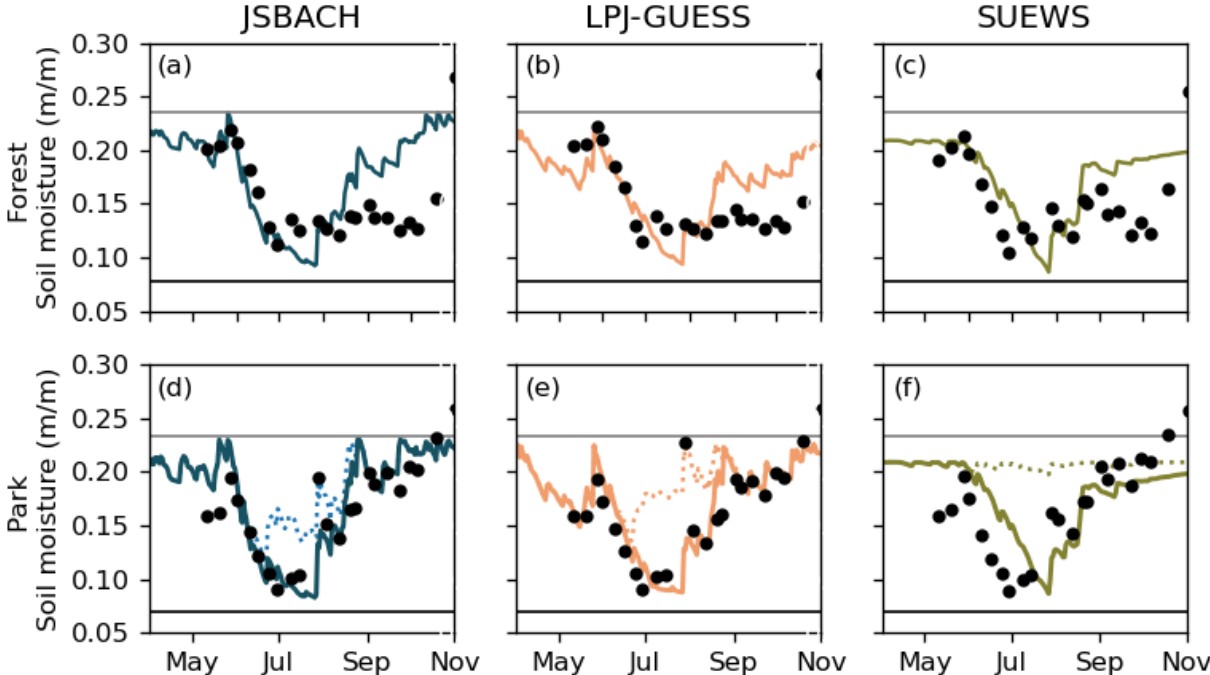

**Figure 3.** Daily mean soil moisture of the root zone estimated by the different models (lines) and observed (dots) in the urban birch forest (upper panels) and the park site with trees (lower panels) from May to October 2021. Solid lines are from non-irrigated simulations and dotted lines from irrigated ones. The horizontal black lines represent the used wilting points and the grey lines the field capacities. The soil moisture was simulated for each model for their specific root zones.

types. The soil hydraulic properties (wilting point and field capacity) are calculated using the Cosby et al. (1984) regression relationships from the soil sand/silt/clay fractions on the measurement sites.

The simulations generally supported the observed soil moisture dynamics in the early season. However, in the latter half of the 2021 season the observed soil moisture in the forest did not increase as rapidly as the simulations predicted (Fig.3 a–c). The Pearson's correlation coefficients were 0.62–0.81 for JSBACH, 0.72–0.78 for LPJ-GUESS, and 0.59–0.63 for SUEWS (Table 2). The soil was moister in 2020 than 2021, and the soil was moist until late autumn, while all the models predicted dryer soil (Fig. S3.) resulting into lower correlations: 0.01–0.38 for JSBACH, -0.11–0.13 for LPJ-GUESS, and 0.26–0.52 for SUEWS.

The extent to which simulated soil moisture increased due to irrigation varied between the models, with the smallest change observed in JSBACH (Fig. 3e) and the most significant in SUEWS (Fig. 3f), which employs its own irrigation scheme. For each model, a value according to its root-zone was calculated from the soil moisture observations.



**Table 2.** Mean Pearson's correlation coefficients between observations and the model estimates. The correlations for soil moisture and sap flow were determined for 2021, leaf area index in 2018–2021, soil temperature, leaf-based photosynthesis (GPP), and soil respiration in 2020–2021, and net ecosystem exchange of $CO_2$ (NEE) in the target area (Fig. 1a) in 2006–2021.

|  |  |  | JSBACH | LPJ-GUESS | SUEWS |
|---|---|---|---|---|---|
| Soil temp | 2020–2021 | park | 0.97 | 0.93 |  |
| Soil moist | 2020 | forest | 0.01 | -0.11 | 0.26 |
|  | 2020 | park | 0.36 | 0.015 | 0.27 |
|  | 2020 | irr park | 0.38 | 0.13 | 0.52 |
|  | 2021 | forest | 0.62 | 0.72 | 0.63 |
|  | 2021 | park | 0.81 | 0.78 | 0.62 |
|  | 2021 | irr park | 0.76 | 0.74 | 0.59 |
| LAI | 2018–2021 | forest | 0.93 | 0.92 | 0.89 |
|  | 2018–2021 | park | 0.89 | 0.92 | 0.89 |
|  | 2018–2021 | lawn | 0.77 | 0.74 | 0.66 |
|  | 2018–2021 | irr lawn | 0.86 | 0.79 | 0.86 |
| Sap flow | 2021 | forest | 0.55 | 0.79 | 0.26 |
|  | 2021 | park | 0.88 | 0.92 | 0.48 |
| GPP | 2020–2021 | park | 0.79 | 0.82 | 0.82 |
| Respiration | 2020–2021 | forest | 0.71 | 0.68 | 0.62 |
|  | 2020–2021 | park | 0.69 | 0.74 | 0.64 |
| NEE | 2006–2021 |  | 0.79 | 0.74 | 0.78 |

## 3.2 Leaf area index (LAI)

All models effectively captured LAI dynamics when compared against satellite observations from 2018 to 2021 (Fig. 4). They
reasonably predicted the timing and magnitude of LAI increment in the different vegetation types studied during spring. In mid-season, LPJ-GUESS and SUEWS generally maintained stable LAI values for trees, whereas JSBACH exhibited a decrease similar to the observations (Fig. 4a–h), due to a small shedding rate applied during the vegetative phase. LAI of lawns exhibited variation based on dry seasons and possible irrigation levels. In dry conditions the net primary production (NPP) in JSBACH is limited by decreasing soil moisture and LAI is decreased. However, low soil moisture does not decrease LAI in SUEWS;
therefore, simulated LAI remained constant for both non-irrigated and irrigated lawns. In contrast, JSBACH and LPJ-GUESS responded to drying conditions in non-irrigated lawns, while the irrigated lawns showed stability in LAI across all models (Fig. 4i and l).





**Figure 4.** Daily one-sided LAI simulated by JSBACH (blue), LPJ-GUESS (orange), and SUEWS (green), and satellite observations (dots) in the studied vegetation types in 2018–2021. The error bars of the observations represent the standard deviation.

Pearson's correlation coefficients indicated differences in model performances with LAI (Table 2). On average, the highest correlation was observed for JSBACH in the urban forest (0.93), for LPJ-GUESS in the park (0.92), for SUEWS and JSBACH in the lawn (0.86), and for JSBACH in the non-irrigated lawn (0.77) (Table 2). Additionally, the mean bias error (MBE) (Table S3) displayed variability between the models. On average, the smallest MBE was associated with JSBACH in the forest (0.03), park (-0.01) and irrigated lawns (0.27), whereas LPJ-GUESS exhibited the lowest MBE for the non-irrigated lawns (-0.01).





### 3.3 Transpiration

Figure 5 shows the correlation of modelled transpiration (from JSBACH and LPJ-GUESS) or evapotranspiration (from SUEWS)
and the sapflow measurements. The correlation coefficient (Table 2) revealed the best agreement for LPJ-GUESS, with r values
of 0.79 for the forest site and 0.92 for the park site. JSBACH also demonstrated good agreement with observations, with r values of 0.55 for the forest site and 0.88 for the park site. In contrast, SUEWS had lower correlation coefficients, primarily due
to the challenge of comparing evapotranspiration instead of transpiration, resulting in values of 0.26 for the forest site and 0.48
for the park site. The comparison between simulated transpiration and sap flow observations was difficult due to the differences
in the units: models simulated transpiration of the whole canopy per ground area whereas the observations described the sap
flow rate per sapwood area. Based on observations, transpiration rates in the urban forest site were lower than those in the
park, as irrigation increased the water availability. Consequently, JSBACH simulated larger transpiration rates for the park than
the urban forest, while LPJ-GUESS maintained transpiration within a similar range in both environments. JSBACH estimated
smaller transpiration rates compared with LPJ-GUESS in the urban forest, whereas in the park site JSBACH estimated larger
transpiration rates. However, as SUEWS simulates evapotranspiration across the whole simulation domain, encompassing both
trees and grasses, direct comparisons with sap flow observations and other models posed challenges. As a result, SUEWS exhibited larger variability and scatter, which was particularly evident at the park site. During peak observation periods, SUEWS
estimated lower values for the urban forest compared to other models, while in the park they fell between estimates from
JSBACH and LPJ-GUESS.

Simulated transpiration varied among the models, which was particularly evident at the forest site from May to June. At
the forest site, SUEWS predicted the lowest rates, while JSBACH predicted the highest values at least for a few weeks (Fig.
S4a). Daily and seasonal variations simulated by JSBACH and LPJ-GUESS closely matched observations during the second
half of the growing period, although showed clear overestimation in early summer. For the park trees, the annual pattern and
day-to-day changes simulated by LPJ-GUESS and JSBACH closely resembled observations in 2021 (Fig. S4b). Furthermore,
the higher evapotranspiration by SUEWS compared to the other models coincided with rainy days, indicating that a substantial
proportion of the evapotranspiration was attributed to evaporation rather than transpiration (not shown).





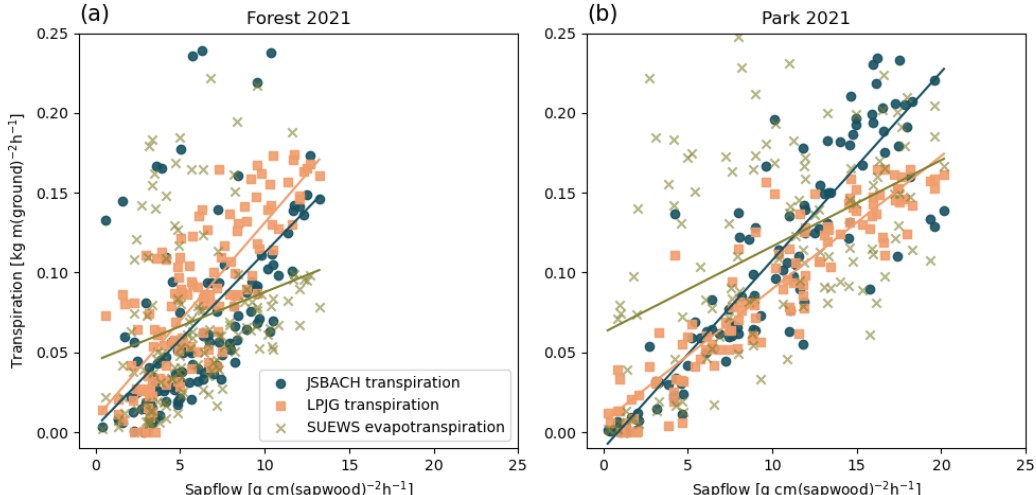

**Figure 5.** Correlation between daily mean sap flow measurements and the modelled transpiration by JSBACH (blue circles), LPJ-GUESS (orange squares), and SUEWS (green crosses) (a) in the urban forest and (b) in the park site in 2021. The measurements are averages over three trees. Note the different units for simulations and observations.

### 3.4 Photosynthesis (GPP)

All model simulations for GPP followed the seasonal dynamics of the daily GPP estimations derived from the manual observations in forest, park, and lawn sites in 2021 (Fig. 6). However, the simulations were per tree-covered land area and observations were mainly per sunlit leaf area, making the comparison of actual magnitudes challenging. GPP by LPJ-GUESS was lower than that of JSBACH and SUEWS in all studied vegetation types, especially in the early season (Fig. 6). JSBACH, LPJ-GUESS, and SUEWS reproduced the observed decrease in GPP during the dry conditions in the urban forest in July 2021 (Fig. 6a). Compared to other models, SUEWS operates differently. While other models simulate different vegetation types separately (urban forest, park, irrigated lawn, and non-irrigated lawn), SUEWS conducts simulations based on the whole vegetation sector and only two separate simulations, a non-irrigated and an irrigated, were run. The non-irrigated simulation of SUEWS was used to represent the GPP of non-irrigated lawns and urban forests, while the irrigated simulation was used for parks and irrigated lawns. However, the daily rates in SUEWS were fairly similar to JSBACH in all studied vegetation types.

At the poorly irrigated lawn plots, the observed GPP decreased during the dry period in July 2021, which JSBACH, LPJ-GUESS, and SUEWS were able to reproduce (Fig. 6c). Although SUEWS also reproduced the impact of the dry period on GPP, it exhibited a much smaller reduction, maintaining relatively higher GPP levels during the dry period. On the adequately irrigated lawn plots, such mid-summer suppression was not observed nor estimated (Fig. 6d) and the modelled GPP was generally higher than in non-irrigated sites. In July 2021 the GPP of the non-irrigated sites were about 30 % of the irrigated ones in JSBACH and LPJ-GUESS and about 70 % of those in SUEWS. On both types of lawns, JSBACH and SUEWS simulated the highest daily rates of photosynthesis and LPJ-GUESS the lowest (Fig. 6).





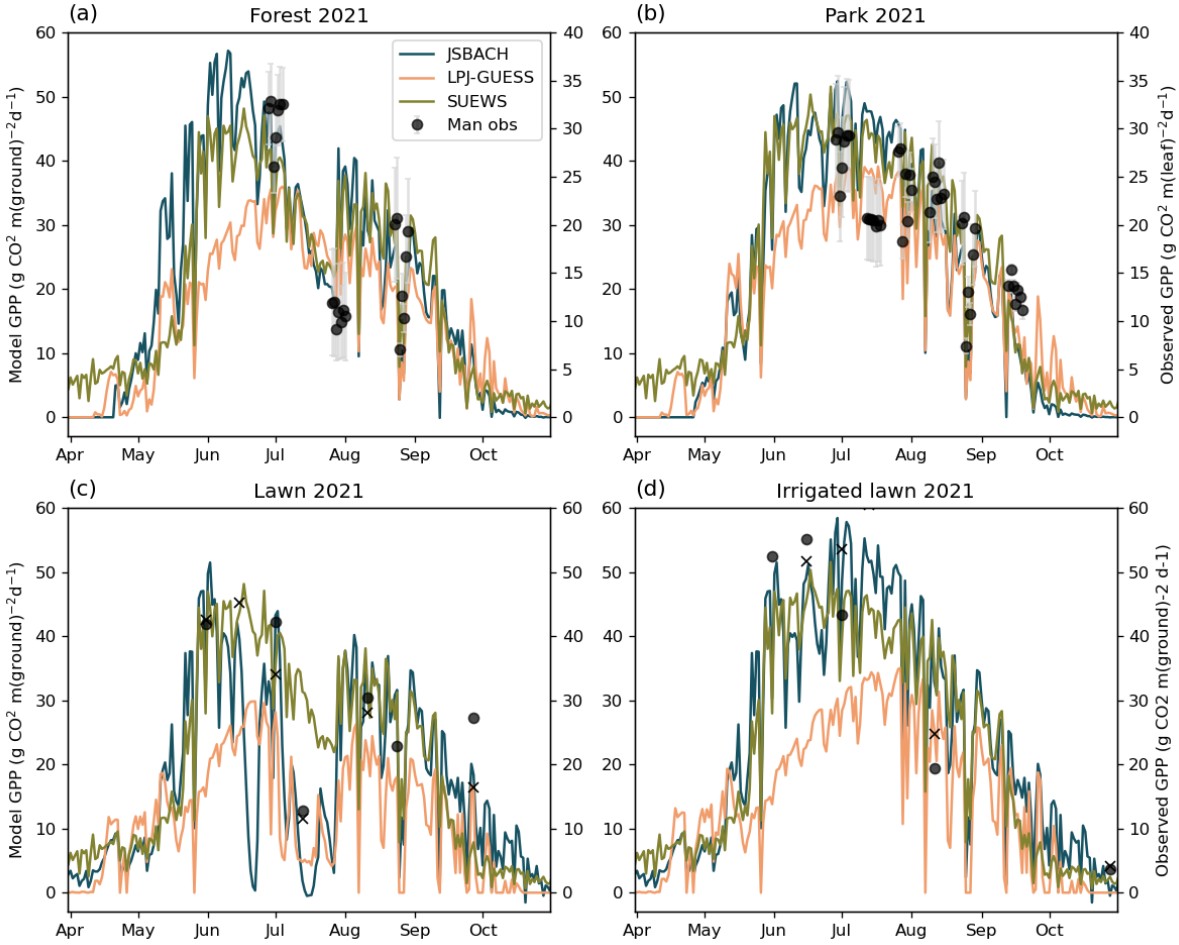

**Figure 6.** Modelled daily photosynthesis (GPP) per ground area and measurement-based leaf-scale estimates of daily photosynthesis in the birch forest (a), the park with linden trees (b), non-irrigated lawn (c), and irrigated lawn (d) in the year 2021. Observations (dots) are averages estimated from the light response curves derived from 9 measurements collected in three different trees. Error bars represent standard deviations of the averages. Note the different units for simulations and observations of forest and park sites.

The GPP derived from the automatic measurements in the park trees provided further support for the ability of the models to simulate seasonal dynamics in GPP. All models were able to estimate autumn senescence in daily GPP observed by the automatic chambers in the park in 2020 (Fig. S5a), resulting in a high correlation between 0.89 and 0.91 (Fig. S6a). In 2021, the models overestimated the early season GPP (Fig. S5b) which caused lowered r values of 0.68–0.76 (Fig. S6b) even though in the latter half of the season the temporal dynamics were again similar between models and observations (Fig. S5b). LPJ-
GUESS and SUEWS showed the highest two-year average r values of 0.82 (Table 2).



## 3.5 Soil respiration

The models displayed reasonably high correlations with mean daily observations of soil respiration under park trees and the urban forest (Table 2). JSBACH simulated just heterotrophic respiration ($R_H$), thus we only examined $R_H$ for both JSBACH and LPJ-GUESS resulting in a difference in the levels between simulated $R_H$ and modelled soil respiration (RE, Fig. 7), but the seasonal dynamics aligned well (Figs. S7 and S8). The $R_H$ by JSBACH was about half of the observed soil respiration, while LPJ-GUESS seemed very low (Figs. 7, S7 and S8).

JSBACH, in particular, demonstrated reasonable correlation with observations in both forest and park sites, achieving r values of 0.69–0.71 (Table 2). Figure 7 shows that the JSBACH simulates comparable $R_H$ values for both non-irrigated forest and irrigated park trees in both years. However, simulations showed variations between years. In 2021, a dry year, both sites exhibited lower respiration values compared to 2020, a normal year. Both sites and years aligned quite well with observations, in 2020 r values were 0.63 for park and 0.77 for forest. However, in 2021 the results were in the opposite order, irrigated simulation results in an r value of 0.76, while the non-irrigated simulation performed worse with a lower r value of 0.65. In contrast, LPJ-GUESS simulated the lowest respiration values among all the models but maintained reasonable r values of 0.68 for non-irrigated and 0.74 for irrigated trees (Table 2). It underestimated soil respiration during both years (Fig. S7). In Fig. 7 it can be seen that modelled $R_H$ is of the same magnitude for both sites, especially in 2020. The difference between years was also small in the LPJ-GUESS simulations. In 2020, SUEWS provided reasonable estimates with r values of 0.65 and 0.66, but its performance in 2021 was not as good, as it tended to overestimate soil respiration during the summer, resulting in slightly lower r values of 0.59 and 0.63.

The analysis revealed seasonal variations and the influence of dry conditions on respiration across forest and park sites. JSBACH and LPJ-GUESS models depicted a reduction in respiration during dry periods (Fig. S7). At the same time, SUEWS showed similar soil respiration values with the irrigated simulation by JSBACH during dry periods in 2021 (Fig. S8). Examining ecosystem respiration across different simulations revealed that SUEWS cannot consider the impact of soil moisture on respiration, unlike the other models. This suggests that SUEWS, initially designed for irrigated low vegetation, may overestimate respiration when applied to non-irrigated lawns.





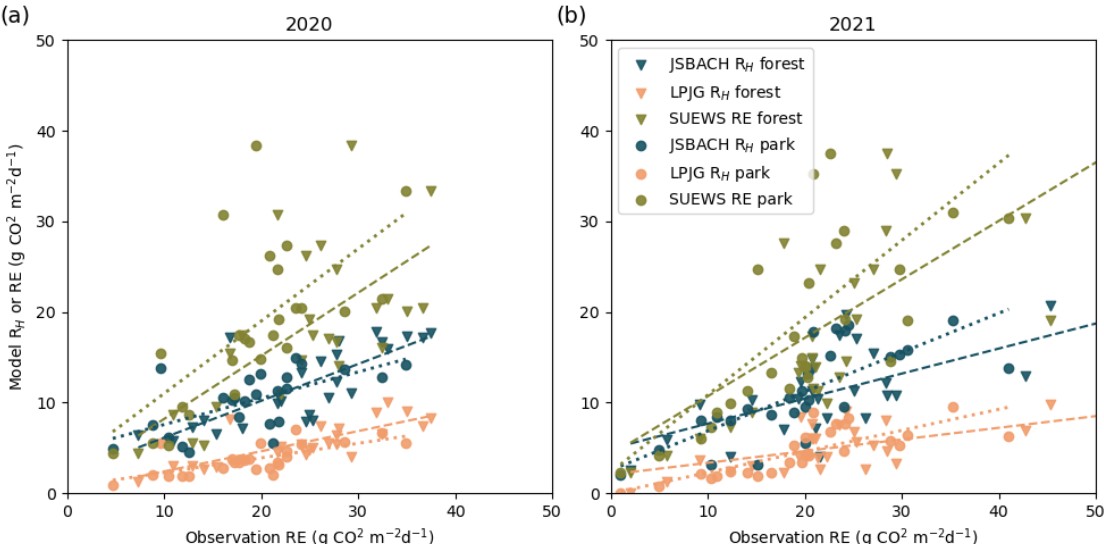

**Figure 7.** Correlation between daily mean soil respiration (RE) by observations and simulated RE or heterotrophic respiration ($R_H$) by the different models for the park tree and urban forest sites in 2020 (a) and 2021 (b). Modelled values are $R_H$ by JSBACH and LPJ-GUESS and RE by SUEWS. Observations are means of manual observations from 8 collars. Dashed lines show the fit between modelled and observed non-irrigated forested sites. Dotted lines are the fits between irrigated park site simulations and observations. SUEWS results are from the irrigated simulation.

### 3.6 Net ecosystem exchange (NEE) over the vegetation sector


Each vegetation type was simulated separately by JSBACH and LPJ-GUESS, and the NEE of the vegetation sector (Fig. 1a) was calculated from these separate NEEs. SUEWS simulated the sector NEE. The simulated results were compared to the estimate of NEE derived from EC measurements. Evaluation of model performance was particularly complicated during the spring recovery due to low measurement coverage (Figs. 8, S9, Table 3). Additionally, the comparison was further complicated

by the uncertain amount of anthropogenic emissions consistently observed. On average, SUEWS estimated the anthropogenic emissions to be around $0.84\,\mathrm{g\,C\,m^{-2}\,day^{-1}}$ (Järvi et al., 2019). Nevertheless, the models were able to roughly simulate the observed range of the daily and seasonal dynamics in the summertime NEE across the diverse target area (Fig. 8). The correlation coefficients between the EC measurements and the different models averaged between 0.74 and 0.79 (Table 2), varying across different years between 0.63–0.88, 0.58–0.85, and 0.64–0.87 for JSBACH, LPJ-GUESS, and SUEWS, respectively (Table 3).

The models displayed varying seasonal patterns in NEE (Fig. S9). On average, SUEWS simulated the highest emissions during the winter and the most substantial sinks during the summer, while the lowest fluxes during both seasons were observed in simulations by LPJ-GUESS. It simulated less seasonal variation in NEE than was observed (Fig. S9a). Even if the midsummer sink was the lowest in LPJ-GUESS, it mostly showed similar annual sinks over the target area (Figs. 8, 9). Nevertheless, the r values for the years 2006–2021 were nearly as high as those of JSBACH and SUEWS (Table 3). Throughout the growing sea-



**Figure 8.** Measured (dots) and modelled daily mean net ecosystem exchange (NEE, $\mathrm{g\,C\,m^{-2}}$) of the target area (Fig. 1) during summer months (May–September) from 2007 to 2021. The last panel shows the average over 2006–2021.

son (Fig. 8, JSBACH and SUEWS provided comparable NEE estimates, although some differences were noticeable. In 2010 and 2020, JSBACH indicated an earlier sink, whereas SUEWS tended to produce larger sinks during autumn compared to EC observations. One of the most prominent distinctions between the models was related to their wintertime (October–April) NEE estimations, with average winter sum values of 160, 61, and 205 $\mathrm{g\,C\,m^{-2}\,year^{-1}}$ for JSBACH, LPJ-GUESS, and SUEWS, respectively (not shown).

From 2006 to 2021 the models demonstrated pronounced annual variation in NEE across the vegetation sector. The average NEE values were -40 ($\pm$34), -44 ($\pm$19), and -52 ($\pm$55) $\mathrm{g\,C\,m^{-2}}$ for JSBACH, LPJ-GUESS, and SUEWS, respectively (Table 3). For the majority of years, the models estimated the vegetation sector to act as a carbon sink. However, there were a few





exceptions. In 2008 and 2017, JSBACH indicated that the sector could act as a small carbon source (2.7 and 0.76 $\mathrm{g\,C\,m^{-2}}$), while SUEWS suggested that in 2010 (27 $\mathrm{g\,C\,m^{-2}}$), 2018 (7.2 $\mathrm{g\,C\,m^{-2}}$), and 2021 (2.2 $\mathrm{g\,C\,m^{-2}}$) the sector could function

as a source. The positive values are small except for SUEWS in 2010. Then also, the summer NEE sum of SUEWS was smaller than average. That was not the case in the years JSBACH simulated a source; then the summer sums were similar to the average value (Table S4). Thus, higher winter emissions played a substantial role in generating the simulated annual source for JSBACH. In contrast, during the source years for SUEWS, the NEE values during the summer, ranging from -149 to -192 $\mathrm{g\,C\,m^{-2}}$, were notably lower than in other years simulated with SUEWS (average -257 $\mathrm{g\,C\,m^{-2}}$), although they were similar

to JSBACH. In contrast, LPJ-GUESS consistently estimated the sector to be a sink, primarily attributed to low wintertime emissions. While the models showed variations in their annual sink strengths relative to each other (Fig. S10), they collectively identified 2010, 2014, 2018, and 2021 as years with the weakest summer sinks (Table S4). Significantly, these years coincided with high irrigation demand (Table 1). Overall, the models demonstrated substantial annual sinks in 2009 and 2015. In 2009 JSBACH and SUEWS achieved high correlation coefficients of 0.84 and 0.85, respectively, and LPJ-GUESS's correlation was

0.70. In 2015, all models achieved high correlations ranging from 0.83 to 0.84, benefiting from more extensive data coverage compared to most other years (Table 3).

### 3.7 Carbon sequestration in different urban vegetation types

According to the simulations, the non-managed urban forest and the irrigated park with linden were on average stronger sinks than the irrigated and non-irrigated lawns (Fig. 9). The mean NEE over 2006–2021 for forested sites were approximately -93

– -92 and -157 – -143 $\mathrm{g\,C\,m^{-2}}$ with JSBACH and LPJ-GUESS, respectively, whereas the mean NEE for lawns were app. -55 – -30 and -15 – -7 $\mathrm{g\,C\,m^{-2}}$, respectively.

During the summer months (May–September) the tree-covered vegetation types were still the strongest sinks, but in JSBACH the lawns were nearly as high. In LPJ-GUESS, the difference between trees and lawns was remarkable. During summer LPJ-GUESS was the smallest sink although it was the largest when it comes to the whole year (Fig. S11).

Irrigation increased the carbon sink of the lawn on average by 84 % and 94 % (25 $\mathrm{g\,C\,m^{-2}}$ and 7 $\mathrm{g\,C\,m^{-2}}$) in JSBACH and LPJ-GUESS simulations, respectively. Irrigation in the park did not show as notable an effect on NEE as in the lawn. JSBACH estimated the sink to increase with the irrigation 22 %, and in the case of LPJ-GUESS the sink the increase was 27 %. Irrigation did not notably affect the simulated inter-annual variation in NEE estimated in the park but decreased the inter-annual variation in the sink of the lawn (Fig.9). Non-irrigated lawn's NEE was +74 $\mathrm{g\,C\,m^{-2}}$ by JSBACH and +64 $\mathrm{g\,C\,m^{-2}}$ by LPJ-GUESS in

2018, which was considered a dry year. In the moist year 2015, the NEE was estimated to be -183 $\mathrm{g\,C\,m^{-2}}$ and -104 $\mathrm{g\,C\,m^{-2}}$ by JSBACH and LPJ-GUESS, respectively.

## 4 Discussion

Quantifying biogenic carbon fluxes within diverse urban areas presents challenges due to the restricted capacity for direct measurements. This task is particularly challenging in urban areas due to the varying microclimate, surface properties, and



**Table 3.** Yearly net ecosystem exchange (NEE, $\mathrm{g\,C\,m^{-2}}$) and Pearson's correlation coefficients between observed and simulated NEE by different models over the diverse urban area (Fig. 1a) and observation data coverage during the year (%) in different years. Negative NEE values indicate a sink of carbon.

|      | JSBACH | | LPJ-GUESS | | SUEWS | | Coverage % |
|------|--------|------|--------|------|---------|------|------------|
|      | NEE    | r    | NEE    | r    | NEE     | r    | Year       |
| 2006 | -20.15 | 0.88 | -63.16 | 0.84 | -113.95 | 0.84 | 12.88 |
| 2007 | -31.56 | 0.80 | -42.82 | 0.74 | -33.68  | 0.71 | 20.27 |
| 2008 | 2.68   | 0.87 | -26.42 | 0.84 | -72.95  | 0.78 | 24.59 |
| 2009 | -102.53| 0.84 | -78.16 | 0.70 | -132.86 | 0.85 | 20.00 |
| 2010 | -44.85 | 0.82 | -45.95 | 0.73 | 27.20   | 0.84 | 19.45 |
| 2011 | -23.01 | 0.74 | -38.90 | 0.74 | -13.34  | 0.76 | 34.52 |
| 2012 | -57.50 | 0.86 | -35.52 | 0.84 | -99.39  | 0.87 | 25.96 |
| 2013 | -49.78 | 0.63 | -64.91 | 0.62 | -6.59   | 0.64 | 22.47 |
| 2014 | -44.22 | 0.78 | -5.22  | 0.77 | -10.99  | 0.76 | 22.74 |
| 2015 | -105.90| 0.83 | -74.96 | 0.83 | -137.45 | 0.84 | 38.63 |
| 2016 | -87.63 | 0.76 | -51.34 | 0.64 | -81.29  | 0.76 | 31.15 |
| 2017 | 0.76   | 0.87 | -27.24 | 0.85 | -20.99  | 0.86 | 27.95 |
| 2018 | -15.37 | 0.64 | -36.86 | 0.58 | 7.16    | 0.70 | 23.56 |
| 2019 | -29.12 | 0.74 | -48.90 | 0.60 | -114.73 | 0.82 | 17.26 |
| 2020 | -19.09 | 0.73 | -23.71 | 0.74 | -23.09  | 0.75 | 31.15 |
| 2021 | -17.22 | 0.82 | -34.77 | 0.80 | 2.15    | 0.76 | 23.90 |
| Ave  | -40.28 | 0.79 | -43.68 | 0.74 | -51.55  | 0.78 | 24.78 |
| STD  | 33.55  | 0.08 | 19.58  | 0.09 | 54.91   | 0.07 | 6.64  |

human management. While measurements offer valuable insights, they alone are insufficient for predicting future climate impacts. Hence, models play a crucial role in complementing measurements and bridging the gap in understanding carbon dynamics. This study assessed the performance of three models for simulating an area in Helsinki utilizing comprehensive observations gathered from diverse urban vegetation types. Subsequently, these models were employed to project the annual carbon sink across various vegetation types under typical weather conditions.

Based on the analysis, all models studied showed the ability to represent most of the variation in the tested parameters, that is, soil moisture, leaf area index, transpiration, photosynthesis, soil respiration, and net ecosystem exchange with default parametrization or with some minor modifications. However, the absolute values were not always simulated correctly, but on the other hand, especially comprehensive tree- and ecosystem-level measurements are difficult to collect and therefore the observations also have shortcomings. Nevertheless, the seasonal dynamics in simulated C fluxes were mainly correct including

declines caused by a dry period in the middle of the season. There were, however, some systematic discrepancies which we will discuss next.





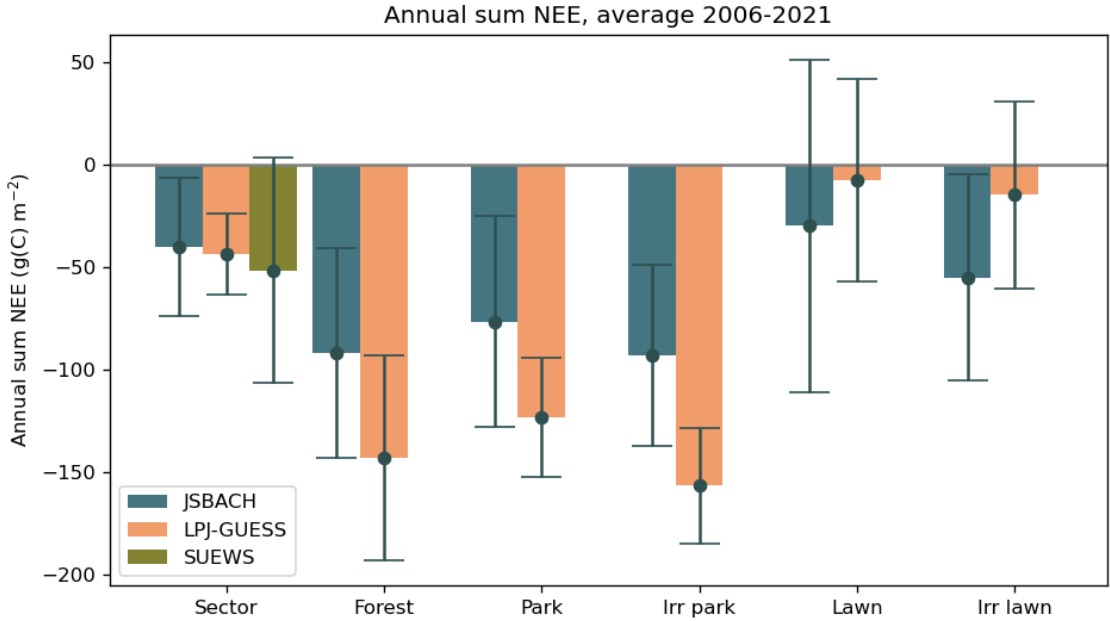

**Figure 9.** Mean carbon sequestration (NEE) in the diverse target area (Fig. 1a) and for the separate vegetation types in the different models over 2006–2021. The error bar represents the standard deviation over the years.

## 4.1 Applicability of the models to study urban C dynamics

JSBACH and LPJ-GUESS successfully simulated soil temperature and moisture, which are key drivers of processes such as organic matter decomposition (Davidson and Janssens, 2006; Chapin et al., 2011). Heterotrophic respiration, associated

with organic matter decomposition, represents a significant carbon flux and the largest natural $CO_2$ emission (Ryan and Law, 2005). Yet, the models tended to overestimate the increase in moisture after the drought period in the urban forest. There are many possible reasons for this. Firstly, measuring soil moisture is challenging (Tarantino et al., 2008; Rasheed et al., 2022), secondly, part of the precipitation is intercepted in the canopy (Dohnal et al., 2014; Kermavnar and Vilhar, 2017) or run-off (Ilvesniemi et al., 2010), and thirdly, re-filling the water storage of trunks might consume more water than estimated. This

is supported by the observations of sap flow, which featured a slightly larger increase than what the models estimated for transpiration/evapotranspiration (Fig. S4). However, this alone is unlikely to explain the discrepancies between observations and the model.

The models successfully simulated the annual cycle of photosynthesis as well as the drought responses on non-irrigated plots, but unfortunately the spring measurements were infrequent. Leaf area index data was more widely available and mainly

supported the simulations of the timing and dynamics of the spring awakening. However, the averaged annual cycle in NEE



indicates that the models predict sinks to increase too early (Figs. 8, S9). The discrepancy between this and the satisfactory LAI estimation may be due to the photosynthetic capacity of plant leaves being weaker during the growth stage compared with mature leaves. Alternatively, the models may underestimate growth respiration and other early season emissions, or the phenology of the dominant vegetation in the target area may differ from the vegetation types observed in detail. At the same time, assessing the timing and the speed of spring recovery in different years from NEE data is also challenging due to both poor data coverage, especially in springtime, and the influence of anthropogenic fluxes. In other ecosystems, EC measurements are commonly gap-filled (Moffat et al., 2007; Vekuri et al., 2023), but in urban areas the precise identification of source areas at each moment and constantly changing anthropogenic emissions pose additional difficulties for the process (Menzer et al., 2015). Therefore, future studies would benefit from high-frequency GPP measurements during the spring period. Setting up automatic chambers for growing vulnerable leaves is challenging without causing significant damage. Therefore, intensive manual measurements could be utilized to assess photosynthetic rates in tree leaves and lawns. Similarly, more frequent measurements would be needed to follow the spring recovery for grasslands.

It is problematic to definitively determine which model performs the best using the simulations over the main source area of NEE measurements, but JSBACH and SUEWS show slightly higher explanatory power than LPJ-GUESS (r=0.79 and 0.78 vs. 0.74). Similarly, the amplitude between winter and summer is more consistent with observations for JSBACH and SUEWS than for LPJ-GUESS (Fig. 8). Again, using urban NEE in model evaluation is challenging due to anthropogenic emissions and low data coverage during a single year. Visually, there are individual years when every model is performing nicely. In general, JSBACH and SUEWS simulate the component fluxes (GPP and soil respiration) similarly and generally predict greater variability in both photosynthesis and natural emissions compared to LPJ-GUESS. In the end, LPJ-GUESS estimates a larger sink for woody vegetation types than JSBACH, but this can be explained by a lower respiration rate relative to photosynthesis. This is in line with Mäki et al. (2022) who studied respiration rates in coniferous forest soils along a latitudinal gradient in Europe and reported underestimated $R_H$ by LPJ-GUESS. Figure S12, illustrating daily cumulative carbon exchange among different models for different years, demonstrates how LPJ-GUESS results in the same annual sink, despite its instantaneous photosynthetic rates being lower than those of other models for both grass and woody vegetation.

## 4.2 Different vegetation types

According to our results from both JSBACH and LPJ-GUESS, the tree-covered vegetation types are higher annual sinks of carbon than the lawns in Helsinki. However, LPJ-GUESS estimated higher sink in tree-covered and lower in lawns compared with those by JSBACH. For the urban forest, the mean annual NEE was -92 and -127 $\mathrm{g\,C\,m}^{-2}$ by JSBACH and LPJ-GUESS which are lower than reported by Luyssaert et al. (2007) who estimated mean net ecosystem productivity (NEP) to be 178 and 311 $\mathrm{g\,C\,m}^{-2}$ for boreal and temperate deciduous forests, respectively. Nowak et al. (2013) estimated annual carbon sequestration to be 280 $\mathrm{g\,C\,m}^{-2}$ on average in urban trees in the United States and Havu et al. (2024) estimated city-level NEE to be -160 $\mathrm{g\,C\,m}^{-2}\,\mathrm{yr}^{-1}$ in Helsinki. Regarding lawns, our annual NEE estimates (-7 – -55 $\mathrm{g\,C\,m}^{-2}$) are also lower than those by Reitz et al. (2021) who used the eddy covariance technique to estimate the annual carbon sequestration of grass to be -131 $\mathrm{g\,C\,m}^{-2}$ in Germany. Our results are more in line with Wohlfahrt et al. (2008) who estimated the NEE of maintained



grass to be -18 g C m$^{-2}$ yr$^{-1}$ in Austria. Thienelt and Anderson (2021) reported NEE of an irrigated lawn to vary from -131 ($\pm$ 24) to -18 ($\pm$ 22) g C m$^{-2}$ yr$^{-1}$ in Colorado. Naturally, as NEE (and NEP) is the small difference in two opposite and much larger fluxes (GPP and RE) these observation-based estimates not only include some uncertainty but also depend on climate and, for example, the soil carbon content and other properties, therefore comparing individual studies without considering all other aspects is complicated.

In previous studies, soil respiration in urban green spaces was reported to vary on average between 0.025 and 0.079 mg C m$^{-2}$ s$^{-1}$ in Helsinki in May–October (Järvi et al., 2012). In Boston, soil respiration was reported to be 0.031 ($\pm$ 0.002) and 0.054 ($\pm$ 0.002 mg C m$^{-2}$ s$^{-1}$) in urban forests and lawns, respectively, during a growing season (Decina et al., 2016). In Quebec, soil respiration was found to average 0.016 mg C m$^{-2}$ s$^{-1}$ on a frequently managed lawn during a growing season (Allaire et al., 2008). In Moscow, soil respiration in urban parks was reported to vary between 0.065 and 0.901 mg C m$^{-2}$ s$^{-1}$

(Sushko et al., 2019), and slightly lower values (0.013–0.139 mg C m$^{-2}$ s$^{-1}$) were reported in Kursk, Russia (Sarzhanov et al., 2017). As the observations show such a wide variation, our measurements easily fall into the range of the literature by being in the scale of 15–30 g CO$_2$ m$^{-2}$ day$^{-1}$, which corresponds to 0.05–0.09 mg C m$^{-2}$ s$^{-1}$. The models resulted in larger maximums, 35–50 g CO$_2$ day$^{-1}$ (0.11–0.16 mg C m$^{-2}$ s$^{-1}$), but still clearly under the highest reported rates. The annual respirations given by our models were 430–730 g C m$^{-2}$ yr$^{-1}$ for irrigated lawn and 320–590 g C m$^{-2}$ yr$^{-1}$ for non-irrigated

lawn. These are also in line with Jasek-Kamińska et al. (2020) who estimated that urban grasslands emit about 424 ($\pm$ 43) g C m$^{-2}$ yr$^{-1}$ in Krakow and with Allaire et al. (2008) who estimated that emissions of frequently maintained lawns in Quebec are annually around 545 g C m$^{-2}$ yr$^{-1}$.

   In a recent study conducted at irrigated lawns in Colorado, NEE was reported to vary with respect to the annual climatic conditions: -131 ($\pm$ 24) on a normal year, and -18 ($\pm$ 22) g C m$^{-2}$ yr$^{-1}$ on a year hit by a severe drought (Thienelt and

Anderson, 2021). Thus, it seems clear that water availability plays a key role in CO$_2$ fluxes in urban green spaces. Irrigation is a common practice to improve both plant vitality and also improving CO$_2$ uptake in urban green spaces, but at the same time it aids the decomposition of organic matter via improved soil moisture. Yet it is still unclear how irrigation influences net C exchange, particularly in urban areas where the soil carbon content is unstable, and thus hardly predictable (Pouyat et al., 2006; Setälä et al., 2016; Ivashchenko et al., 2019; Sushko et al., 2019; Cambou et al., 2021). We showed that dry periods during the

summer remarkably decrease the LAI of the non-irrigated lawn in JSBACH and LPJ-GUESS simulations and observations, and that the GPP of an irrigated lawn is higher than that of a non-irrigated one. On average, irrigation increased the carbon sequestration by 17 and 33 g C m$^{-2}$ yr$^{-1}$ (22 % and 27 %) according to JSBACH and LPJ-GUESS, respectively, at tree-covered sites. On lawns, irrigation increases the carbon sequestration by 25 and 7 g C m$^{-2}$ yr$^{-1}$ (84 % and 94 %) according to these two models, respectively. Such comparisons with different vegetation types is not possible using SUEWS but the annual

NEE at the target area (Fig. 1a) turned into a small carbon source (+2 g C m$^{-2}$ yr$^{-1}$) from a sink (-51 g C m$^{-2}$ yr$^{-1}$) when the irrigation was turned off in the simulations. Irrigation increased the GPP in the target area by 18 %, 16 %, and 6 % in JSBACH, LPJ-GUESS, and SUEWS, respectively. At the same time, the respiration of irrigated soil was larger than non-irrigated. In a typical year, here 2020, the difference was quite small with only 7 % and 13 % more respiration due to irrigation in JSBACH

0

 





and LPJ-GUESS, respectively, but in 2021, a dry year, the irrigation increased the respiration by 42 % and 53 % in JSBACH
and LPJ-GUESS.

These results conflict with Livesley et al. (2010) who found that in Melbourne irrigation itself did not significantly affect the
$CO_2$ fluxes. However, Thienelt and Anderson (2021) reported that, when irrigation was stopped for a short period during warm
days on urban lawns in Colorado, soil moisture and LAI were highly impacted, and as a result NEE became positive during
this short time. This result was also demonstrated in Helsinki by Trémeau et al. (2024), where GPP and respiration reached low
values during drought events on a non-irrigated lawn, but remained more stable on an irrigated lawn. In Los Angeles under a
dry climate, it has also been found that the maximum $CO_2$ uptakes occur during the peak of irrigation in summer (Miller et al.,
2020). On the other hand, it was calculated by Jasek-Kamińska et al. (2020) that $CO_2$ emissions were at their maximum when
the soil moisture was between 27 % and 32 %, but GPP was not measured in that study. Zirkle et al. (2011) found in a modelling
study that irrigation could improve the carbon sequestration by $10\,\mathrm{g\,C\,m^{-2}\,yr^{-1}}$. Thus, the amount of irrigation needs to be
optimized to find the right trade-off between photosynthetic uptake and the $CO_2$ released by the increased decomposition rate
of soil organic matter.

### 4.3  Year-to-year variation

All models simulated high year-to-year variation in annual net ecosystem exchange (Table 3) which was driven by the variation
in weather. SUEWS showed the largest variation in NEE at the target area (-137–+27 $\mathrm{g\,C\,m^{-2}}$) whereas LPJ-GUESS resulted
in the lowest variation (-78– -5 $\mathrm{g\,C\,m^{-2}}$). According to all models, the sinks in 2009 and 2015 were the highest, accompanied
by comparably high EC data coverage and high correlation coefficients. These years were not considered especially warm,
cold, moist, or dry (Table S2), indicating that none of these features seem to favour net carbon sequestration in the studied
conditions.

To some extent, temperature and moisture increase both net primary production and the decomposition of soil organic
matter but the relationships are not linear. According to a meta-analysis (Rustad et al., 2000), warmer temperatures increased
soil respiration remarkably, and the response was stronger in forests than in grassland ecosystems. A study by Meineke et al.
(2016) investigating the effects of warming on urban tree function and growth in NC, USA, found that warming reduced tree
growth and carbon sequestration. They concluded that the potential of urban trees to act as a carbon sink will decrease in the
future. In addition, an extended period of drought limits the plant's water and ion intake due to impaired root-soil contact, root
shrinkage, and anatomical deformations (Lipiec et al., 2013). As NEE is a sum of the input and output, a detailed analysis
of the flux components would improve our understanding of the role of different weather years and the effect of increasing
temperatures and increasing possibility of extended drought in urban vegetation in northern cities.

In general, all models estimated higher NEE for the target area during the whole study period than during the years of
measurements (2018–2021), which highlights that measurements collected over a couple of years might not represent the
common variability in weather and carbon fluxes of the area of interest. Investing in long-term measurement campaigns or,
as here, utilizing models trained with the data allows us to estimate finer variability, which is not possible with measurements
collected over a couple of years.



## 4.4 Suggestions for improvement

The modeling of seasonally varying LAI exhibits differences among the models, necessitating some adjustment of vegetation
phenology in all instances. In the case of SUEWS, this adjustment entails fine-tuning the dependence of growing or senescence
degree days on temperature, guided by satellite measurements. In contrast, LPJ-GUESS accounts for the results of the previous
growing season to determine the maximum LAI on the subsequent one, which is in some cases reasonable but not in heavily
managed ecosystem types such as lawns. Here, the LAI dynamics of JSBACH were adjusted based on Sentinel-2 data, and for
lawns the critical temperature, which needs to be exceeded to allow growth, was also adjusted to meet the observations. Given
the sensitivity of carbon uptake to LAI development, accurately capturing seasonal dynamics becomes imperative. Therefore,
it is recommended that the phenological patterns in different cities should be tested in further use of the model, or at least that
the vegetation dynamics should be adapted to the climate of the respective city.

Furthermore, the temporal resolution employed in this study was hourly for JSBACH and SUEWS, while it was daily
for LPJ-GUESS in v4. However, a more recent version of LPJ-GUESS (LPJ-GUESS/LSMv1.0, Martín Belda et al., 2022)
enables the simulation of diurnal exchanges of energy, water, and carbon between the land ecosystem and the atmosphere.
This modification holds the potential to improve the simulation of carbon fluxes. It also has a 9-layer soil column, which
improves the water holding of the soil. However, this updated model version is not yet widely adopted and therefore, the
currently more common older version was used in this study. This study used measured soil composition data (clay and sand
content). According to the soil composition values in the original soil map of LPJ-GUESS, the soil in Kumpula would have
been more clayey, that is, better at retaining water. With such soil, LPJ-GUESS would have produced up to 20 % higher GPP
and respiration values. Furthermore, NEE would have indicated up to 10 % larger carbon sink (Fig. S13).

The significance of this comparative study lies in the mutual learning opportunities that different models provide one another.
The urban land surface model, SUEWS, incorporates various urban elements crucial for modeling carbon sequestration in
cities, addressing factors such as the urban heat island effect and integrating an irrigation module. As highlighted previously,
the role of irrigation in simulating carbon fluxes is of considerable importance. In SUEWS, irrigation is incorporated into the
model, specifically accounting for how trees and grasses uptake carbon and its correlation with water stress. However, a notable
limitation arises when comparing SUEWS to other models, as respiration rates are calculated without accounting for soil water
content. This limitation restricts the applicability of SUEWS for more detailed soil respiration analysis. Additionally, SUEWS
comprises only one soil layer beneath the surface, drawing all its moisture storage from this single layer. In contrast, other
models incorporate at least two soil layers, enabling plants to extract water from each at different rates. This feature facilitates
a more dynamic and realistic simulation of water availability for plant transpiration.

Conversely, the other models, JSBACH and LPJ-GUESS, which do not incorporate some urban aspects, must be evaluated
independently. The impact of elevated air temperatures on vegetation phenology can be parameterized, and irrigation can be
estimated to supplement precipitation amounts. Without considering these additional aspects, the models would not be suitable
for urban areas.





## 5 Conclusion

As cities worldwide strive towards carbon neutrality, understanding the complex dynamics of urban vegetation and its impact on carbon fluxes is essential. While comprehensive measurement of diversity of urban vegetation types and growing conditions remains a challenge, the tested models exhibit strengths in simulating seasonal and year-to-year changes in carbon fluxes

and their drivers, such as leaf area, soil moisture and soil temperature. However, validating absolute levels of net ecosystem exchange of mature trees is hindered by observational limitations, which usually measure just single sunlit leaves. Moreover, our findings underscore the significant influence of irrigation on carbon fluxes, highlighting the importance of incorporating this factor into models such as JSBACH and LPJ-GUESS. As soil moisture affects both the decomposition of soil organic matter and the vitality of trees, controlled experiments could optimize irrigation to support carbon sequestration and model

development. Additionally, improving SUEWS respiration estimates by integrating the effects of soil moisture could further enhance the accuracy of carbon flux modelling in urban environments. As research in this field continues to evolve, addressing these model-specific developmental needs and refining our understanding of urban carbon dynamics will be important for sustainable urban planning and climate mitigation efforts.

*Data availability.* The measurement data used in the study can be accessed and downloaded at following sites:

Manual GPP of lawn: https://doi.org/10.23728/fmi-b2share.920c1e5f08a74a6d9dfcb3a08cfc6734

Manual GPP and sapflow of trees: https://doi.org/10.5281/zenodo.7525319

Soil temperature, moisture and respiration: https://doi.org/10.57707/fmi-b2share.f7ba414bfd3642168ac38a95835b06bc

LAI: https://doi.org/10.5281/zenodo.5993292

NEE: https://doi.org/10.57707/fmi-b2share.e638f63a3e6f45eb890e964726154964

Automatic GPP: https://doi.org/10.57707/fmi-b2share.840b8a856abf43e18b3fbb329eed5305

Model results: https://doi.org/10.57707/fmi-b2share.0cb5e547dd2f48da89c1b690604dd3d0

Driver data: Finnish Meteorological Institute open data: Timeseries API, opendata.fmi.fi/timeseries, last access: 16 March 2022.

SMEAR data: https://smear-backend.rahtiapp.fi/search/timeseries/csv

*Author contributions.* LT, LB, MH and LK conceptualized the study. LT, LB and MH executed all model simulations. EK, JS, JT, ON, JA, LJ and LK contributed to the data curation. LT performed the formal analysis. LT and EK contributed to the visualization. MH, LJ and LK acquired funding. LJ and LK supervised the research planning. LT, LB, and MH contributed to the software development. LT, LB, MH and LK contributed to the preparation of the original draft. All authors contributed to the writing - review & editing.

*Competing interests.* The authors declare that they have no conflict of interest.



*Acknowledgements.*  This research was funded by the Strategic Research Council working under the Academy of Finland (grant #335201, 335204), the Academy of Finland (grant #325549, 358257), the ACCC Flagship program of the Academy of Finland (grant #337552, 337549) and Tiina and Antti Herlin Foundation.

We greatly acknowledge Jarkko Mäntylä, Juho Aalto, Heikki Laakso, Pasi Kolari, Pasi Aalto and Eki Siivola for the technical assistance and Mikael Lindholm and the whole staff in Kumpula Botanic Garden for various assistance and favourable attitude. Yasmin Frühauf, Anni
Karvonen, Pinja Rauhamäki and Olivia Kuuri-Riutta are acknowledged for their help in the field measurements and Quentin Bell for help with the language.



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
