# Peer review of "Carbon sequestration in different urban vegetation types in Southern Finland"

_EGUsphere, 2024_

## Author Comment (AC1)

**REFEREE 1**

General comments

The study employs three models of different detail to analyze carbon fluxes at three sites of different green infrastructure. The models are initialized, parameterized, and partly evaluated with measured data. From the description, the evaluation seems to be a bit poor in cases, particularly regarding the grassland site, which makes the comparison of different plant types rather difficult. To my feeling, the drought stress, which might be an important driver for carbon dynamics is not convincingly reflected in the models. Nevertheless, the simulations seem to be able to represent the overall carbon fluxes for the tree sites and results might also give an indication about the grassland dynamics. Still, I would be happier when the discussion would be more carefully formulated. Despite deficiencies, I think that the analysis has some merit and may serve for an improved measurement setup as well as model developments.

Dear Reviewer, thank you for your careful review and feedback to improve the quality of the manuscript! We have considered all comments and revised the manuscript accordingly. We do recognize some shortcomings with the grassland site which we address in detail in the point-by-point responses below together with the description of the revisions.

We agree that the drought stress would deserve more attention but after careful consideration, we concluded that the material and discussion in this paper are already quite extensive and instead, we will address the drought in a separate paper. Also, we will conduct controlled irrigation experiments in the area during the next few growing seasons, which will provide more data to improve the models further. In this paper, the drought response in the models regarding GPP seems to be reasonable, while the models disagree on the response in grass LAI, and to some extent also in the soil moisture. In JSBACH and LPJ-GUESS the drought response is driven by both LAI and soil moisture, while in SUEWS the GPP response is from the soil moisture changes.

We have revised the discussion, removed any unnecessary repetition and reviewed the overall formulation, in addition to addressing the detailed comments given.

Below you can find the responses provided point by point.

**Abstract**

P1L1: 'in sequestration potentials' instead 'in the sinks'

Thank you, done.

P1L6: delete 'momentary'

Thank you, done.

P1L9: replace 'concerning mature trees and that in urban areas' with 'because' (and adjust to 'includes' in the next line)

We revised that to read as follows, "However, validation of absolute fluxes proved challenging because of observational limitations, particularly for mature trees, and the fact that net ecosystem exchange measurements in urban areas include some anthropogenic emissions."

P1L12: do you mean 'for the same vegetation type' instead 'across a studied urban vegetation area'?

Here, we do mean the area that is the sector (Fig. 1a) which consists of multiple vegetation types. The following sentence gives more information. Basically, the differences in different green area types were evened out on the area. For clarification, we changed the wording to 'over a studied urban vegetation sector'.

P1L16: replace 'key' by 'single'

Thank you, done.

Introduction

P2L27: Some more recent reviews would be e.g. Ferrini et al. 2020 or Cuthbert et al. 2022

Thanks for the suggestion. We added new citations.

"Cities worldwide are actively engaged in climate change mitigation efforts, formulating strategies to decrease their anthropogenic emissions (Rosenzweig et al., 2010; Reckien et al., 2014; Mitchell et al., 2022; Ferrini et al., 2020)…"

".. as well as improving  human health and well-being  (Wolf et al., 2020; Cuthbert et al., 2022)."

 P2L29ff: I see what you would like to say. Still, I think it would be favorable if the argumentation would be clearer that carbon fluxes are on the one hand indicators for other ecosystem services and on the other hand easy targets for physiological-oriented models. And that therefore the models need to be employed in order to also judge the development of ecosystem services.

 Thanks for the suggestion! We have revised this accordingly but have not yet mentioned the modelling part in this paragraph, as we introduce the general need for modelling in the following paragraph. The revised sentence now reads:

'As a result, understanding and quantifying carbon storage and fluxes in urban green areas and the impact of management practices on carbon is necessary to mitigate climate change through smart design. As changes in carbon fluxes represent changes in the overall ecosystem functioning, those can also be used to indicate some other ecosystem services provided.'

The following, revised paragraph begins as follows,

'The heterogeneity of urban green spaces, coupled with the year-to-year variability in weather patterns, poses challenges for empirically estimating carbon sequestration. Consequently, physiological-oriented ecosystem models are needed to quantify both the present extent and the future development of carbon sequestration and at the same time, potentially some other ecosystem services.'

P2L54: replace 'utilized' with 'used to estimate carbon sequestration'

Thank you, done.

P3L62: replace 'validation' by 'evaluation' (also check throughout the manuscript)

Thank you, this is changed throughout the revised manuscript.

**Methods**

P10L250: What is meant with 'momentary photosynthesis' is a simulated assimilation rate that is based on half-hourly climate data which are supposed to be representative for this period, correct? TER and NEE are calculated for the same periods but are not called 'momentary'. Please proof me wrong but I think this term inconsistent, not exactly correct, and superfluous.

Thank you, you are right. In the original manuscript, we used momentary photosynthesis for the 30-min average photosynthesis to distinguish that from mean or cumulative daily photosynthesis which were compared with the model simulations. We have standardized the terminology in the revised manuscript and we are no longer mentioning momentary photosynthesis but wherever necessary, included a clarification on the time period it represents.

P10L264: Is it correct that the models have some light absorption algorithms that consider that parts of the tree canopies are always shaded? Could you indicated that such a process is applied in all models to account for self-shading? Otherwise photosynthesis rates would be of course too high.

In JSBACH the photosynthetically active radiation (PAR) in the canopy is described by the two-stream approximation (Dickinson, 1983), which describes the budget of the diffuse radiation in the canopy. One stream describes the downward diffuse radiation, while the other describes the upward diffuse radiation. The direct radiation is attenuated exponentially in the canopy. Scattering of the direct beam produces diffuse radiation, by reflection of transmission. Part of the diffuse radiation is absorbed by the leaves, but a fraction is scattered back, which then contributes to the respective other stream. The canopy is split into several layers of equal depth (fraction of LAI) to account for the non-uniform distribution of Rubisco in the canopy. In this way the fAPAR can be calculated taking into account the self-shading of the canopy.

In LPJ-GUESS, radiative transfer follows Beer's law (Monsi and Saeki, 1953). The canopy is divided into vertical layers, each of which absorbs a portion of the PAR transmitted by the layer above. The PAR absorbed by each layer is then distributed among plant cohorts based on their respective share of the leaf area index (LAI) in that layer. This means that taller cohorts have access to more PAR and cast shade over the lower layers of the canopy.

In SUEWS, the photosynthesis parameters were derived from CO2 flux data observed by eddy covariance (EC) measurements over the vegetation sector from 2006 to 2011 (Järvi et al., 2019). The model does not account for self-shading; instead, it considers average radiation conditions measured at the tower and relates them to the average CO2 flux. Unlike other models, SUEWS focuses on simulating the overall CO2 flux from urban vegetation, without delving into detailed representations of individual trees.

We updated the model descriptions regarding self-shading.

> P11L272: What does 'set to a fixed value of 415ppm' mean. Is it possible to regulate the CO2 concentration in such a small measurement chamber? Did you check the ambient CO2 concentration? As it is known that CO2 concentrations in cities are often way higher than that, it seems necessary to confirm that this is not the case or explain how the models are able to address this issue.

It is indeed possible to regulate and maintain the concentration in the chamber close to the set value with Walz. Naturally, there is some variation, especially after a change in conditions, which in this case was a change in PAR intensity. In practice, the measurement included 16 different light intensities, automatically changing from 1500 down to 0 $\mu mol\ m^{-2}\ s^{-1}$ over 43 minutes, so that at each light intensity there was enough time to adjust the input of $CO_2$ into the chamber to meet the changed rate of assimilation. In any case, the $CO_2$ exchange rate is calculated using the measured $CO_2$ concentration inside the chamber and not the set value.

Truly, $CO_2$ concentrations are high in urban areas, especially during the night, which however is not an important period for photosynthetic uptake even in the Nordic countries, where the nights are bright, but the light intensities are still low. Here, we chose the set concentration to roughly represent the daytime ambient concentrations in the mixed boundary layer observed in the area. Below are the measured mean daily $CO_2$ concentrations between 10-16 local time during the main summer months in 2020-2021 at a nearby station in Helsinki. It shows that the $CO_2$ concentrations during these hours were close to 405-415 so in our opinion, the upper end of this range is acceptable to account for possible increased $CO_2$ concentrations in the morning and late evening hours.

We added the information also in the sentence in question which now reads, as follows:

'During each measurement, $CO_2$ concentration was set to a fixed value of 415 ppm representing the typical daytime $CO_2$ concentration in the area, but the temperature was not set to any value and was following the ambient conditions.'

[Figure]

*Figure 1: CO2 concentrations measured at FMI building roof in Kumpula from June to August in 2020-2021.*

P11L285: Soil respiration was measured at 8 plots at each of two sites? How were they distributed over the sites to ensure representativeness? What about the grassland site – are the models evaluated with it before/ at other places?

Soil respiration was, indeed, measured at 8 points both in the park (under the linden trees) and in the urban forest. The measurement points were situated along two almost parallel transects with 4 points on each so that there were, on average, 4 meters between the transects and 3 meters between the measurement points on each transect. As described in Karvinen et al. (2024), "the aim of the [measurement point] selection was to capture the spatial variation within each measurement site by ensuring there is enough distance between the single measurement points and having some of them be located closer to trees than others, some closer to the edge of the green space than others, and so on". To make it easier for the interested reader to find this information without adding more length to the methods section of this paper, we have changed the last sentence of this paragraph to "The chamber system is described in detail by Pumpanen et al. (2015) and Karvinen et al. (2024) and the measurement point selection and flux calculation by Karvinen et al. (2024)."

As described in the manuscript, we derived light response measurements of NEE on the lawns using manual chambers. That included also one measurement in full darkness, i.e., TER. We, however, did not compare this value to model simulations in lawns mainly because there is a large diurnal variation in soil and surface temperature in unshaded areas without tree coverage and therefore, it is difficult to generalise the one TER measurement taken in varying hour and ambient conditions to daily value at which LPJ-GUESS operates. However, we have already tested the TER simulations by JSBACH in different type of urban grassland in Tremeau et al. (2024). In the revised manuscript, we have included a short discussion on the uncertainty in the annual NEE values caused by the missing TER testing and reference to Tremeau et al. (2024) in the first paragraph in the section.

P12L303: In order to directly compare measurements and simulations, it would be favorable, if you could derive mm m-2 h/day-1 values from the sapflow measurements. Could this be done?

We agree that it would be better to compare the transpiration derived from sapflow rates with the simulated transpiration. However, the transformation would require the sapwood area of the study trees, which we do not know for sure. Nevertheless, if we assume a sapwood area of 433.8 cm$^2$ (Ahongshangbam et al. 2023) and a tree density of 330 trees/ha for the linden trees (Park) and for the birch trees (Forest), sap wood area of 349.7 cm$^2$ and 500 trees/ha, respectively, we can compare the transpiration estimates per tree (kg/h). The comparison is presented below in Fig. 2. It shows that the model estimates are in the lower range estimated by the standard deviation in the sap flow rates. The calculation was incorporated into the revised M&M section (sap flow observations) and the result into the Results section. However, it was not deemed necessary to really emphasise this, given that the sapwood areas are merely estimates and that, for instance, drilling the study trees to estimate the actual sapwood area is not allowed.

[Figure]

*Figure 2 Hourly sapflow (shaded areas) and simulated transpiration (lines) per tree (kg/h) estimated from the sap flow rates and JSBACH and LPJ-GUESS simulations for the forest site (a) and for the park (b) trees. The shaded area represents the standard deviation in the individual sap flow measurement rates.*

**Results**

P13L333: If you have simulations and data for 2020 too, please show them also in the graphs.

We have simulations and observations for 2020, and they are shown in the supplement (Fig. S3). To make it clearer, we added a sentence about it, and now it reads: "Soil moisture in the urban forest and park sites for 2020 is shown in Fig. S3."

P16L366ff: If I understand correctly, it is not possible to judge if simulated transpiration rates are higher or lower than reality. The comparison as it is in Fig. 5 is only indicating that the dynamics are possibly related. Thus, formulations need to be chosen very carefully. Please revise.

That is correct. In the original version, we had only compared different habitats and different models to each other, and not the magnitude of the models' transpiration rates to the magnitude of the observations. We clearly checked the whole section and made one small clarification and one sentence to clarify that further: " Thus, the correlation coefficients indicate the similarity of seasonal dynamics between models and observations." After that, we added the sap flow based transpiration estimates as described see earlier.

P19L407ff: Does LPJ Guess calculate soil respiration? In this case, not only RH (to compare with JSBACH) but also the full value should be shown in Fig. 7 and be discussed. I would also suggest to put in a 1:1 line.

Thanks for the suggestions! LPJ-GUESS calculates also root-, sap- and growth respiration. We added RE_lpjg = $R_H$ +RA(root)+RA(growth) and a 1:1 line to the Fig. 7 and RE_lpjg also to the Fig. S7. Also, short discussion of the new results was added in the revised manuscript.

The text now reads: "The models displayed reasonably high correlations with mean daily observations of soil respiration under park trees and the urban forest (Table 2). JSBACH simulated heterotrophic respiration ($R_H$), thus we examined $R_H$ for both JSBACH and LPJ-GUESS for comparing the models, and RE from LPJ-GUESS and SUEWS. $R_H$ comparison resulted in a difference in the levels between simulated R_H and modelled soil respiration (RE, Fig. 7), but the seasonal dynamics aligned

well (Figs. S7 and S8). The $R_H$ by JSBACH was about half of the observed soil respiration, while both $R_H$ and RE by LPJ-GUESS seemed very low (Figs. 7, S7 and S8)."

LPJ-GUESS simulated the lowest respiration values among all the models but maintained reasonable r values of 0.68 for non-irrigated and 0.74 for irrigated trees in case of $R_H$ (Table 2) and better r values of 0.84 for non-irrigated and 0.80 for irrigated trees in case of RE.

[Figure]

*Figure 3. New version of Fig 5, where RE from LPJ-GUESS is added.*

> P20L442: This is not very well formulated. While single peaks of emission in summer are indeed very high with LPJ Guess, I daresay that the vegetation period is still a larger sink than the winter time.
>
> P20L443: similar over the target area – what area? Or over the years?

We meant that although the LPJ-GUESS sink during the summer is small (shown in Figure 8), still the annual NEE calculated for the whole vegetation sector in Kumpula (Figure 9) is the same size as the other models. We'll change the sentence: "When comparing all the models, the midsummer sink was the lowest in LPJ-GUESS (Fig. 8) but the annual sinks over the target area were similar in all the models (Fig. 9)."

> P21L445: parenthesis?

Thank you, done.

**Discussion**

> P22L83ff: The first paragraph is more or less a repetition of parts in the introduction. Consider deleting.

We agree that there was some repetition, and we deleted the whole first paragraph and also made some small modifications in the second one.

> P23L493ff: replace 'absolute values' by 'measurements', 'simulated correctly' by 'realistically represented' and 'mainly correct' by 'reasonably close to measurements' or similar. I am also struggling with the expression of 'systematic discrepancies. Do you mean that the model structure and output sometimes doesn't match the scale and targets of measurements? Consider rephrasing.

We changed as suggested. With discrepancies we actually meant mismatches and revised that to 'systematic disagreement.'

P24L504ff: Don't the models consider interception? Your assumption implies that canopy interception is underestimated although the leaf area is more or less correctly met. This doesn't seem very likely. Also, I cannot imagine that stem water refilling might be responsible for the long-term water deficit. On the other hand, I could imagine that soil properties, in particular preferential flows (in addition to runoff) might represent a considerable uncertainty.

We agree and you are correct that the models include already interception. We revised the section as follows,

'Firstly, measuring soil moisture is challenging (Tarantino et al., 2008; Rasheed et al., 2022), and secondly, part of the precipitation is lost as run-off (Ilvesniemi et al., 2010) or transported to lower soil layers by preferential flows. In addition, re-filling the water storage of trunks might consume more water than estimated which is supported by the observations of sap flow, which featured a slightly larger increase than what the models estimated for transpiration/evapotranspiration (Fig. S4). However, this alone is unlikely to explain the discrepancies between observations and the model. Some of the precipitation is naturally intercepted in the canopy (Dohnal et al., 2014; Kermavnar and Vilhar, 2017) but this process is already included in the models.'

P25L514: replace 'from vegetation types observed in detail' by 'because of a different composition or structure of the actual vegetation'

Thank you, done.

P25L519: replace 'future studies would benefit from' with 'measurements would particularly been improved by'

We have revised the sentence to be clearer, and now it reads as follows: 'Therefore, testing would be improved by high-frequency GPP measurements, particularly in the spring period.'

P25L520: replace 'for' by 'to measure'

Thank you, done.

P25L523: replace 'problematic to definitively determine' by 'not possible to decide' or similar

Thank you, done.

P25L526: replace 'is challenging' by 'has its drawbacks' or similar. In addition, it might be argued while some targets e.g. drought related phenology was best met in LPJ, others, such as the carbon exchange during spring, was better represented by the other models. And thus, every model has its strong- and weak points.

We did the replacement and added a sentence of LPJ-GUESS's drought phenology. Now it reads: "However, each model has its strengths and weaknesses as the drought-related phenology was best met by LPJ-GUESS."

P27L593ff: Did you check radiation? Years might be particularly good if cloudiness is low and radiation might be high.

Thanks for the suggestion! We calculated accumulated summer radiation and also the number of days when the radiation was lower than average. These numbers were added to Table S2 in the Supplement. However, neither of these parameters helps explain the large NEE values of certain years. We changed the sentence like:

"According to all models, the sinks in 2009 and 2015 were the highest, accompanied by comparably high EC data coverage and high correlation coefficients. These years were not considered especially warm, cold, moist, dry sunny or cloudy (Table S2), indicating that none of these features seem to favour net carbon sequestration in the studied conditions."

*Table 1. A new version of Table S2 with accumulated summer radiation and the number of cloudy days.*

**Table S2.** Growing degree days with a base temperature of 5°C (GDD5, degree days), accumulated precipitation during summer (May–August, mm), accumulated short wave radiation during summer months ($kWm^{-2}$), maximum temperature (Max T, °C), the number of days when precipitation is below 1.4 mm (Dry days) and the number of days when daily shortwave radiation is lower than average in Kumpula during 2006–2021.

|      | GDD5 | Summer precipitation | Summer radiation | Max T | Dry days | Cloudy days |
|------|------|----------------------|------------------|-------|----------|-------------|
| 2006 | 1846 | 87  | 30.05 | 25 | 108 | 32 |
| 2007 | 1656 | 320 | 27.14 | 23 | 81  | 54 |
| 2008 | 1538 | 220 | 26.46 | 22 | 87  | 62 |
| 2009 | 1539 | 325 | 26.99 | 23 | 86  | 57 |
| 2010 | 1745 | 211 | 26.95 | 26 | 94  | 57 |
| 2011 | 1863 | 310 | 27.70 | 25 | 90  | 48 |
| 2012 | 1538 | 258 | 27.61 | 23 | 87  | 50 |
| 2013 | 1781 | 212 | 27.19 | 24 | 96  | 52 |
| 2014 | 1710 | 280 | 25.28 | 26 | 81  | 59 |
| 2015 | 1518 | 236 | 26.75 | 22 | 89  | 60 |
| 2016 | 1636 | 270 | 26.65 | 21 | 92  | 52 |
| 2017 | 1389 | 211 | 26.54 | 19 | 95  | 61 |
| 2018 | 1964 | 144 | 30.48 | 27 | 105 | 34 |
| 2019 | 1679 | 215 | 28.81 | 25 | 100 | 43 |
| 2020 | 1720 | 296 | 29.27 | 24 | 94  | 40 |
| 2021 | 1777 | 293 | 28.08 | 27 | 94  | 48 |
| Ave  | 1681 | 243 | 27622 | 24 | 92  | 51 |
| STD  | 151  | 65  | 1399  | 2  | 8   | 9  |

P28L620ff: Consider rephrasing. What do you actually recommend? An empirical adjustment of phenological pattern for cities? A better initialization based on measurements? A better representation of varying environmental impacts such drought into the models?

We revised these sentences to be more clear. 'Therefore, it is recommended that the phenological patterns are considered in the model initialisation before further use in other cities. '

References

Ahongshangbam, J., Kulmala, L., Soininen, J., Frühauf, Y., Karvinen, E., Salmon, Y., Lintunen, A., Karvonen, A., and Järvi, L.: Sap flow and leaf gas exchange response to a drought and heatwave in urban green spaces in a Nordic city, Biogeosciences, 20, 4455–4475, https://doi.org/10.5194/bg-20-4455-2023, 2023.

Dickinson, R.E., Land surface processes and climate — Surface albedos and energy balance. Advances in Geophysics 25 (1983) 305-353.

Järvi, L., Havu, M., Ward, H. C., Bellucco, V., McFadden, J. P., Toivonen, T., Heikinheimo, V., Kolari, P., Riikonen, A., and Grimmond, C. S. B.: Spatial modeling of local-scale biogenic and anthropogenic carbon dioxide emissions in Helsinki, Journal of Geophysical Research: Atmospheres, 124, 8363–8384, 2019.

Karvinen, E., Backman, L., Järvi, L., and Kulmala, L.: Soil respiration across a variety of tree-covered urban green spaces in Helsinki, Finland, EGUsphere, 2024, 1–35, https://doi.org/10.5194/egusphere-2023-3031, 2024.

Monsi, M. and Saeki, T.: On the Factor Light in Plant Communities and its Importance for Matter Production, Jpn. J. Bot., 14, 22–52, 1953.

Trémeau, J., Olascoaga, B., Backman, L., Karvinen, E., Vekuri, H., and Kulmala, L.: Lawns and meadows in urban green space –a comparison from perspectives of greenhouse gases, drought resilience and plant functional types, Biogeosciences, 21, 949–972, https://doi.org/10.5194/bg-21-949-2024, 2024.

---

## Author Comment (AC2)

**REFEREE 2**

> The article compares 3 models to analyze carbon sequestration in urban vegetation (irrigated and non-irrigated lawns, park trees, and urban forests). The study considered various parameters such as soil moisture and temperature, sap flow, leaf area index, momentary photosynthesis, soil respiration, and net ecosystem exchange. Evaluation of all these parameters and their presentation makes this article a bit complicated. Repetition at many places in the Discussion makes it too lengthy.

Dear Reviewer, thank you for your feedback! We have considered all comments and revised the manuscript accordingly, have carefully revised the Discussion to avoid repetition and excessive length. Below you can find the responses provided point by point below.

> The Abstract sufficiently represents all aspects of the research work.

> The introduction is properly written with clear aims and objectives. In some places ( P2L28-28), a few unrelated topics may be removed.

As suggested, we removed the air quality and recreation from the list of ecosystem services provided by urban green spaces.

> Materials and Methods is somewhat lengthy. It should be precise and easy to understand.

We agree that Materials and Methods is lengthy, but there were so many different study sites, unique observations and models together with the irrigation schemes, that it is challenging to describe all clearly but shortly without losing replicability. We read the section carefully and deleted some sentences for example in the model description where some of the original information given was not relevant for the study.

> Observation represents all aspects of the study. In P10L250 is Momentary photosynthesis GPP? Make it clear.

Indeed, we mean the same thing with GPP and photosynthesis. In the original manuscript, we used the term *momentary photosynthesis* to make a clear difference between the 30 min average and daily mean or cumulative sum. However, we have revised the terminology based on the comments by #R1 and we do not use momentary photosynthesis any longer.

> The results of the study are clearly presented. In P13L338 it is written "**The soil was moister in 2020 than 2021",** but data for 2020 is not graphically presented as to compare with 2021.

The figure for 2020 is in the Supplement. To make it clearer, we added a sentence about it, and now it reads: " Soil moisture in the urban forest and park sites for 2020 is shown in Fig. S3.

> The discussion contains repetition at places.

We have carefully reviewed the discussion section and removed the first paragraph, for example, as it was not relevant and included some repetition.

> References are sufficiently provided.

---

## Author Response (AR2)

Dear authors,

I agree with the reviewer of this second round of peer review, who also reviewed the original version, that the quality of the manuscript has clearly improved, but that there are still a number of issues raised by the reviewer that need to be addressed before I can definitively recommend your work for publication. The reviewer is still not fully satisfied with some of your responses and how you have addressed the reviewer's criticisms in the revised version. Your argumentation could be improved and made more stringent, while the length of the text could be reduced by removing sentences that have no real meaning. See below for specific points. Kind regards,
 Nicolas Brüggemann

Associate Editor

Dear editor Nicolas Brüggemann

Thank you for your feedback and for the opportunity to revise our manuscript! We appreciate the acknowledgment of the improvements made so far. We agree with the reviewer's new comments and have addressed them thoroughly in the revised paper. See below the detailed responses to each comment. We have further strengthened our argumentation and ensured our revisions directly address the criticisms. During this process, we also deleted sentences with low value to enhance clarity and shorten the discussion.  In addition, the second sentence of 4.1 has been relocated to the beginning of the section, as this proved to be a more logical arrangement, and the final sentence of the first paragraph in 4.3 has undergone minor revisions.

We are committed to refining the manuscript as needed and look forward to any additional guidance.

Sincerely, on behalf of all authors,

Laura Thölix

Reviewer's comments

L310ff: In contrast to the description and the nice figure 2 in response to my previous comments, the changes made in the text are a bit irritating. Besides from wording issues ('The details of … are described in detail', 'Furthermore' instead of 'Therefore', …) the explanation is unclear. How can you transform data (sap flow records) into 'estimates of transpiration' (referring to which area?) by multiplying them with 'values for sapwood'

that seem to come from literature (but the references are not given in the literature list)? Revise, and possibly provide the resulting figures in the supplementary.

Thank you for your insightful comments! We agree with your observations and have improved the wording in the relevant section to address the issues mentioned. We've also clarified our method for calculating transpiration and added the missing references to the reference list. In the main analysis we compare the seasonal dynamics of sap flow to modeled transpiration. In addition, we also present a comparison of transpiration per tree where the observed sap flow rate is converted into transpiration. The description of the methods was unclear, especially regarding the conversion into transpiration. The revised methods description reads as follows,

> *First, we compared the sap flow rates and the model estimates of transpiration to analyse seasonal dynamics. Second, the sap flow rates were transformed into estimates of whole tree transpiration by multiplying the rates with species-specific values for sapwood area from the literature, namely 349.7 cm² for the birches (Zapater et al., 2013) and 433.8 cm² for park trees (Leuzinger et al., 2010).*

The comparison figure of sap flow driven and model estimates of whole-tree transpiration (Figure S5) is referred to in the first paragraph of subsection 3.3.

L526ff: Also, although better, the discussion of possible 'disagreements' (why not deviations?) is still not very convincing. In particular the impact of a variable trunk water storage should be much too small to be significant. That there is interception is also not a reason for a deviation, except it is explained why the models might do it wrong. Instead, I would mention that the soil water supply in the models is likely different to what real trees experience and perhaps already here mention that the phenology (absolute LAI, temporal and spatial distribution) might be critical to fit measurements to models.

We agree that deviation is more suitable word than disagreement and changed that in the revised MS. We also totally removed the speculation on the trunk water storage and the sentence regarding interception. Instead, we added the suggested topic on model testing and development needs. Now the revised discussion regarding the soil moisture deviation reads, as follows

> *The models had some difficulty in reproducing the temporal dynamics of soil moisture, e.g., the increase following periods of droughts in the urban forest was overestimated in the models. Several factors could lead to discrepancies between model results and observations. Measuring soil moisture is inherently challenging*

*(Tarantino et al., 2008; Rasheed et al., 2022). Additionally, the soil water supply as represented in models is likely to differ from actual conditions. This difference can be attributed to factors such as root depth and soil texture, but also to the phenology. In addition, local variations in the precipitation may not be correctly captured in the forcing data. Some part of the precipitation is lost as runoff (Ilvesniemi et al., 2010) or transported to deeper soil layers through preferential flow pathways, while the soil is assumed to be homogeneous in the models. Addressing these issues is essential to improve the accuracy of soil moisture simulations in future model developments.*

L576ff: The additional explanation about uncertainties in the gas exchange of lawns is certainly important. However, wording is not very good and argumentation is inconsistent. For example, if JSBACH has shown to reasonably estimate (gas exchange? Temperature? Water content?) dynamics of lawns, why is it still a problem that needs to be discussed here? In addition, what is a 'soil media' and why does it influence the organic matter? Do you mean that your initialization of soil properties which are important for flux estimations is uncertain? The last sentence in this paragraph ('Therefore, …') is superfluous and only sounds as an excuse for not having an idea where the deviations are coming from.

Thank you for your notes! Upon reviewing the text, we noticed several structural issues and ambiguities ourselves too. First, we have clarified the explanation regarding the model evaluations made in the previous study. Second, we aimed to convey that there are no specific soil properties, such as carbon content, in urban lawns and it is not thus meaningful to generalize lawn TER based on sparse observations. These soils are not in equilibrium with the primary production of the vegetation as something else may have grown there previously, or the soil may have been relocated. Additionally, commercial substrates (~soil media) with high organic matter and nutrient content are often added when establishing lawns affecting the observed TER. Now, we revised the pointed section to read, as follows

*In our study, we evaluated the dynamics of GPP on lawns but not the TER, as the momentary measurement of TER are difficult to scale up to a daily level. This is because in open areas such as lawns, changes in radiation can cause significant changes in soil temperature leading up to changes also in TER. This naturally causes uncertainty in the estimated TER and NEE but Trémeau et al. (2024) showed that JSBACH can estimate the seasonal dynamics and absolute level of TER in irrigated and non-irrigated lawns in Helsinki. However, heterotrophic respiration at different lawns depends on the quality and quantity of organic matter in the soil, which in turn*

*depends on the history of the soil and possible earlier soil amendments such as mulch. Therefore, soil properties and heterotrophic respiration may vary spatially in urban areas without a clear link to vegetation types, as the carbon cycle is rarely in a steady state yet. Without case-specific information on soil carbon pools, model initialisation will be uncertain.*

We removed the criticized, last sentence as it clearly added no information to the topic.

> L635ff: It is true that forest soils show stronger response to soil warming than grasslands (Rustad et al. 2001). Probably because of their higher organic matter content. However, you need to consider that higher temperatures will heat up grassland soils much more than well shaded forests. Therefore, the same air temperature rise will probably affect forest soils less than grassland soils. And although you are correct that Meineke et al. indeed found a reduction of aboveground carbon storage under urban warming, this doesn't seem to apply for soils as pointed out by Rustad. Therefore, argumentation need to be better and more carefully formulated. Apart from that, before demanding a more detailed analysis of flux components, authors should specify the uncertainties of the current measurements and perhaps reflect on some quite comprehensive papers on this topic (e.g. Ryan 2023, Zahn et al. 2022, Ueyama et al. 2026).

The pointed section was indeed quite weak and clumsy in places. Rustad et al. has been partly misquoted in relation to grasslands, probably because several authors have revised the text and the quote has moved out of its original context. The whole paragraph has been thoroughly improved with additional references and strengthened arguments. Furthermore, a short discussion on EC general reliability, gap-filling, partitioning, and related issues has been added at the end. During the revision, we also removed some sentences that have no real value. Now it reads as follows,

*Increased temperatures are considered to increase soil respiration (Rustad et al., 2000), but a local study showed that increasing soil temperature had less effect than irrigation on heterotrophic respiration in urban tree-covered environments (Karvinen et al., 2024), highlighting the important role of soil moisture also in the north. In subtropical climates warming can also reduce urban tree growth and carbon sequestration (Meineke et al., 2016) but previous article in the hemiboreal study city indicated that temperatures that can locally be considered as extremely high seemed to favour tree photosynthesis (Ahongshangbam et al., 2023).*

*As NEE is a sum of the input and output, a detailed analysis of the flux components would improve our understanding of the role of different weather years and the effect*

*of increasing temperatures and increasing possibility of extended drought in urban vegetation in northern cities. However, gapfilling of eddy covariance C flux data always requires caution (Vekuri et al., 2023; Zahn et al., 2022), and since fluxes in urban areas include diurnal and otherwise varying amounts of anthropogenic sources (Järvi et al. 2012; Ueyama and Ando, 2016), flux partitioning is challenging. It should also be noted that EC measurements may underestimate some of the component fluxes (Ryan, 2023).*

L660: Be careful with your formulation. Phenology might be important to consider in models but not as an initialized value but as a dynamic process that considers the variation of budburst date with spring temperature rise as well as a possible early leaf senescence with increasing drought events. In this context, you might also point out that the drought stress component is not well covered in the model and needs to be better considered (even if you would like to make another paper from this problem).

We agree and changed the formulation from initialization to testing. We also pointed out that the drought issue requires further testing. Now the section reads as follows,

*Therefore, it is recommended that the phenological patterns are tested before further use of the models in other cities. Although the drought response in the models appeared reasonable in terms of GPP, the precise description of thresholds and responses to different drought intensities should be further tested, especially to serve future scenarios.*

L659: I would like to remark that it would be more specific to talk about different scales between measurements and models, rather than limited observations, that make an evaluation difficult.

We agree. We revised the sentence in the conclusion to be as follows,

*However, evaluating absolute levels of net ecosystem exchange in mature trees is hindered by the different scale of observations, which usually focus on single sunlit leaves.*

The second sentence of the discussions section was revised, as follows,

*However, the measurements were not always realistically represented, but on the other hand, comprehensive measurements at tree and ecosystem level are difficult to collect and are therefore at a different scale to the models.*

The issue in the abstract was revised, as follows,

*However, the validation of absolute level of modelled fluxes proved difficult due to differences in the scale of the observations, particularly for mature trees, and the fact that net ecosystem exchange measurements in urban areas include some anthropogenic emissions.*

References mentioned by the reviewer

Ryan, M. G.: The enduring mystery of differences between eddy covariance and biometric measurements for ecosystem respiration and net carbon storage in forests, New Phytol., 239, 2060-2063, https//doi.org/10.1111/nph.19105, 2023.

Rustad, L., Campbell, J., Marion, G., Norby, R., Mitchell, M., Hartley, A., Cornelissen, J., and Gurevitch, J.: A meta-analysis of the response of soil respiration, net nitrogen mineralization, and aboveground plant growth to experimental ecosystem warming, Oecologia, 126, 543-562, https//doi.org/10.1007/s004420000544, 2001.

Ueyama, M., and Ando, T.: Diurnal, weekly, seasonal, and spatial variabilities in carbon dioxide flux in different urban landscapes in Sakai, Japan, Atmos. Chem. Phys., 16, 14727-14740, https//doi.org/10.5194/acp-16-14727-2016, 2016.

Zahn, E., Bou-Zeid, E., Good, S. P., Katul, G. G., Thomas, C. K., Ghannam, K., Smith, J. A., Chamecki, M., Dias, N. L., Fuentes, J. D., Alfieri, J. G., Kwon, H., Caylor, K. K., Gao, Z., Soderberg, K., Bambach, N. E., Hipps, L. E., Prueger, J. H., and Kustas, W. P.: Direct partitioning of eddy-covariance water and carbon dioxide fluxes into ground and plant components, Agric. Forest Meteorol., 315, 108790, https//doi.org/10.1016/j.agrformet.2021.108790, 2022.